EMBO
Molecular Medicine

# Impaired Complex I dysregulates neural/glial precursors and corpus callosum development revealing postnatal defects in Leigh syndrome mice

Sahitya Ranjan Biswas[1], Porter L Tomsick [2], Colin Kelly[1], Brooke A Lester[2], Julia P Milner[2], Sara N Henry[3], Yairis Soto[2], Samantha Brindley[2], Nicole DeFoor[2], Paul D Morton [3✉] & Alicia M Pickrell [2✉]

## Abstract

**Leigh syndrome (LS) is a complex, genetic mitochondrial disorder defined by neurodegenerative phenotypes with pediatric manifestation. However, recent clinical studies report behavioral phenotypes in human LS patients that are more reminiscent of neurodevelopmental delays. To determine if disruptions in epochs of rapid brain growth during infancy precede the hallmark brain lesions that arise during childhood, we evaluated neural and glial precursor cellular dynamics in a mouse model of LS. Loss of Complex I significantly impacted neural stem cell proliferation, neuronal and oligodendroglial progeny, lineage progression, and displayed overt differences in specific brain regions across postnatal development. Our findings show that these disruptions in all categories occur specifically within the subventricular zone and corpus callosum prior to the age when these mice experience neurodegeneration. Given that LS is considered a neurodegenerative disease, we propose that there are neurodevelopmental signatures predating classic diagnosis in LS.**

**Keywords** Leigh Syndrome; Postnatal Neurogenesis; Neural Stem Cells; Subventricular Zone; Corpus Callosum
**Subject Category** Neuroscience

## Introduction

Leigh syndrome (LS) is a severe and prevalent inherited primary mitochondrial disorder affecting children, characterized by progressive neurodegeneration and early mortality (DiMauro and De Vivo, 1996; Lake et al, 2016). To date, mutations in more than 110 nuclear or mitochondrial genes have been found to be associated with the disease, most of which affect assembly subunits or core components of the five complexes responsible for oxidative phosphorylation (OXPHOS) (Lake et al, 2016; McCormick et al, 2023). Consequently, the prevailing

hypothesis regarding disease pathogenesis is that a dysfunctional electron transport chain places increased metabolic stress on high-energy-demanding neurons, ultimately leading to their degeneration (Kayser et al, 2016; Lake et al, 2015). This neuronal loss is reflected in the hallmark lesions observed in different parts of the brain, including the olfactory bulb, brainstem, cerebellum, basal ganglia, midbrain, pons and thalamus, which also account for the phenotypes of the disease and respiratory distress leading to mortality (Ardissone et al, 2021; Goncalves et al, 2020; Kartikasalwah and Lh, 2010).

LS symptomology has been associated with neurodegeneration, where the loss of previously acquired skills in these pediatric patients has long been considered the most prominent feature of the disease (Lake et al, 2016). However, recent follow-up studies in large collective cohorts of LS patients reported that delays in achieving early developmental milestones - such as head movement, rolling, independent ambulation, sitting without support - are often the most common feature, emerging prior to the onset of regressive phenotypes correlating with a poorer disease trajectory (Tinker et al, 2022; Zilber et al, 2023). Most research focuses on understanding the neurodegenerative and neuroinflammatory aspects of the disease. But considering that genetic mutations are inherited, and recent studies indicate that neurodevelopmental defects may occur prior to the onset of neurodegenerative lesions, a more thorough evaluation at the cellular level of the brain, prior to neuron loss, is warranted.

To understand whether key developmental programs underlying early life neurological milestones are affected in LS, we utilized the NDUFS4 knockout (KO) mouse model, a well-established preclinical model of LS characterized by impaired mitochondrial Complex I activity (Kruse et al, 2008; Quintana et al, 2010). This model closely replicates the neuropathological and motor symptoms observed in human patients, including characteristic brain lesions, ataxia, hypotonia, and respiratory failure (Quintana et al, 2012). Moreover, Complex I deficiency remains the most frequent cause of LS, accounting for about one-third of all cases (Rahman, 2025).

NDUFS4 is an accessory subunit of Complex I that is essential for the assembly, stability, and function of this complex that cycles NADH to NAD$^+$, transferring electrons to ubiquinone, generating

[1]Translational Biology, Medicine, and Health Graduate Program, Virginia Polytechnic Institute and State University, Roanoke, VA 24016, USA. [2]School of Neuroscience, Virginia Polytechnic Institute and State University, Blacksburg, VA 24061, USA. [3]Department of Biomedical Sciences and Pathobiology, Virginia-Maryland College of Veterinary Medicine, Virginia Polytechnic Institute and State University, Blacksburg, VA 24061, USA. ✉E-mail: pmorton@vt.edu; alicia.pickrell@vt.edu

the proton gradient for ATP generation (Scacco et al, 2003; Ugalde et al, 2004). Autosomal recessive mutations in *NDUFS4* have been reviewed extensively (van de Wal et al, 2022). From a mechanistic standpoint, the loss of NDUFS4 results in the loss of another subunit NDUFA12 with an overcompensation of NDUFAF2 or NDUFAF6; however, these subunits are unable to compensate for the loss of NDUFA12, resulting in an immature and unstable complex (Adjobo-Hermans et al, 2020; Yin et al, 2024). LS patients within this population harboring these mutations tend to display severe forms of the disease, with over half of these children in one study reporting neurodevelopmental phenotypes (Budde et al, 2003; Lamont et al, 2017; Ortigoza-Escobar et al, 2016).

Using this model, we identified defects in neural stem and progenitor cell proliferation, a preference for quiescence within the stem cell pool, and a decline in neural progenitors undergoing lineage progression into immature neurons during early postnatal development in the subventricular zone (SVZ). In addition, we found evidence of significant declines in gliogenesis, particularly within the oligodendrocyte lineage, paired with a reduced capacity for myelination of the corpus callosum. Together, our findings indicate a selective vulnerability of neural and oligodendrocyte precursors, resulting in delays in maturation and an under-developed corpus callosum prior to the neurodegenerative onset in a LS mouse model. In addition, our transcriptomic data indicate diverse changes amongst many cell populations comprising the neural and glial lineages at early stages of development when there is overlap in neurogenesis and gliogenesis, offering aid in uncoupling the two, along with insights into neonatal contributions to the array of developmental delays exhibited in LS patients.

# Results

## Gross neuroanatomical deficits are notable prior to disease onset in NDUFS4 KO mice

Patients with Leigh syndrome are typically diagnosed following the onset of symptoms related to psychomotor decline, which coincide with neurodegenerative lesions found on MRI (Ruhoy and Saneto, 2014). Previous evaluation of NDUFS4 KO mice characterized signs of neurodegeneration appearing at postnatal day 37 (P37) (Kruse et al, 2008; Quintana et al, 2010; Quintana et al, 2012). To investigate potential early indicators of impaired brain development, we assessed gross brain anatomy that preceded this time point on postnatal days 14, 24, and 30. At these time points, both WT and NDUFS4 KO mice were able to maintain body weight/growth, but gains in brain weight lagged significantly (Fig. 1A,B). Congruent with differences in gross brain weight, we observed a notable reduction in hemispheric width in NDUFS4 KO mice, particularly at P14 (Fig. 1C,D). These data warrant further investigation as they indicate that neurodevelopmental delays precede the documented neurodegeneration in this NDUFS4 KO model of LS.

## Reduced neural stem progenitor populations and impaired lineage progression in the SVZ of NDUFS4 KO mice

We next decided to investigate the cell populations that reside in the SVZ, as it remains postnatally active in rodents and higher-order species (Alvarez-Buylla and Garcia-Verdugo, 2002; Kornack

and Rakic, 2001; Lim and Alvarez-Buylla, 2016; Sohn et al, 2015). To examine cellular changes in the neurogenic pool of the SVZ, we performed immunohistochemistry (IHC) using SOX2 and dou-blecortin (DCX) to delineate neural stem/progenitor cells (NSPCs) and neuroblasts, respectively, at 3 postnatal time points preceding neurodegenerative onset. To assess regional differences within the SVZ, we sampled 5 regions on the coronal plane spanning the frontal lobe corresponding with Bregma areas $+1.53$ to $+0.33$ mm (Fig. EV1A). At P14, we observed a significant reduction in the density of $SOX2^+$ cells in the NDUFS4 KO, indicating a decrease in NSPC numbers; however, these differences were absent at later time points (Fig. 2A,B). At this age, we found the more caudal regions of the SVZ accounted for a significant majority of the reduction in NSPC numbers (Fig. EV1B), which was absent at later timepoints (Fig. EV1C,D). We next evaluated lineage progression as $SOX2^+$ NSPCs upregulate DCX in transition to neuroblasts. At P14, we found no differences in the number of $SOX2^+ DCX^+$ cells nor in the percentage ($SOX2^+ DCX^+/SOX2^+$) of NSPCs undergoing transition to the neuronal lineage; however, there was a significant reduction as early as P24, which persisted into P30 in the SVZ of NDUFS4 KOs (Fig. 2C–E). This significant reduction was accounted for in the second most rostral area (Bregma $+1.23$) assessed (Fig. EV1E,F). We next evaluated neuroblasts and found a significant reduction in the number of $DCX^+$ cells at P14 which was not sustained at the later ages assessed (Fig. 2F). When evaluating the rostrocaudal distribution of the SVZ, we found that most of this difference was within the more rostral regions assessed (Fig. EV1G–I); whereas disruption in lineage progression is evident at later ages, which may indicate NSPCs are shifting toward a less active phenotype or diverging from the neuronal lineage.

Together, these results suggest a biphasic disruption in neurogenesis in complex I-deficient mice. At P14, there were fewer neural stem/progenitor cells (Fig. 2A,B) leading to a reduced generation of neuroblasts (Fig. 2F), which may in part account for the maturational delays in brain growth exhibited (Fig. 1B–D). The reciprocal relationship between reduced NSPCs in the more caudal domains and reduced numbers of neuroblasts in the rostral domains of the SVZ at the earliest age assessed (P14) may also be indicative of reduced tangential migration, owing to fewer numbers of NSPCs generating neuroblasts (Fig. EV1B,G). Considering that mitochondrial disease has genetic origins, we expect that these findings are mainly governed by intrinsic defects in the NSPC pool rather than extrinsic factors such as microenvironmental cues. However, it is possible that other cells in the brain that lack NDUFS4 contribute indirectly.

## Decreased neurogenesis in NDUFS4 KO

While a reduction in NSPC numbers within the early postnatal SVZ may be indicative of cell death, our longitudinal findings demonstrate normal numbers of NSPCs at the expense of a decline in lineage progression (Figs. EV1B,E–G and 2A,C,F); therefore, we reasoned that these early losses may be due to disruptions in cellular genesis. We next assessed NSPC proliferation within the SVZ of WT and NDUFS KO animals following immunohistochemistry at P14. We found a significant reduction in general cell proliferation within the SVZ following quantification of $Ki67^+$ cells (Fig. 3A). Regardless of genotype, the average number of proliferating NSPCs ($SOX2^+Ki67^+$) represented 55–57% of cells under-going mitosis ($Ki67^+$) within the SVZ (Fig. 3B). However, we found a

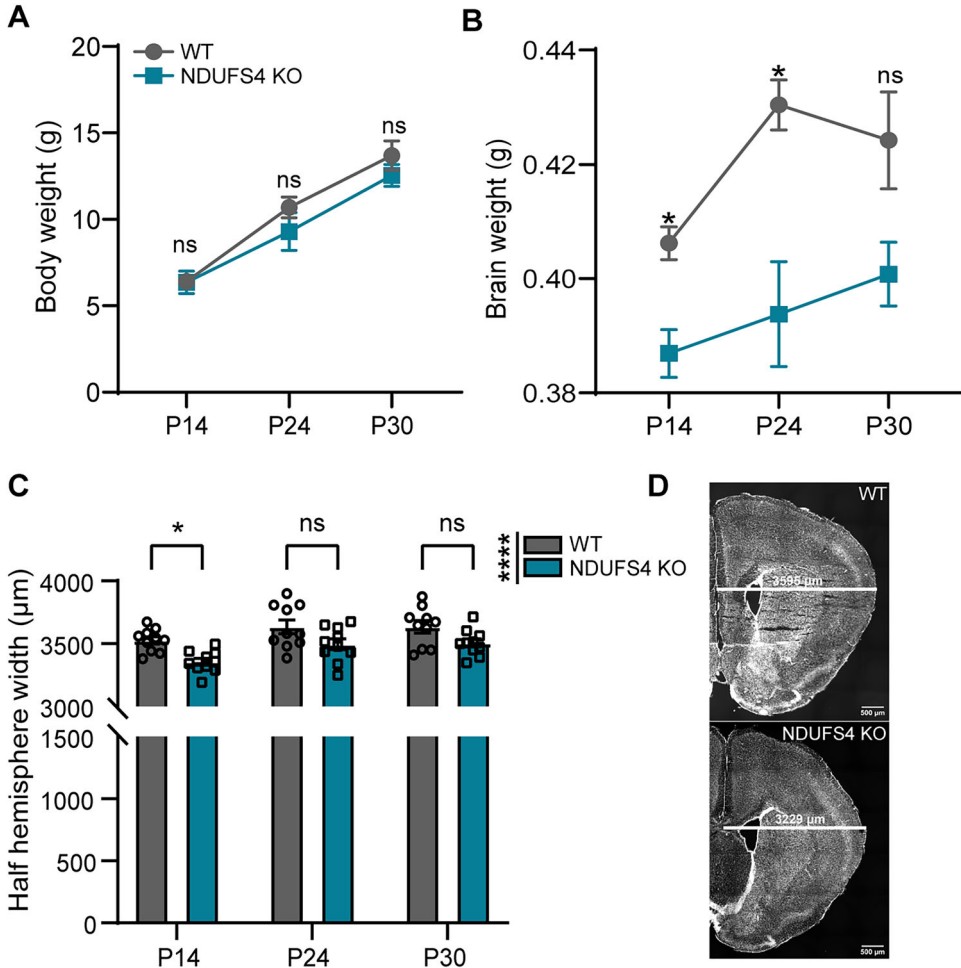

**Figure 1. NDUFS4 KO mice exhibit early postnatal alterations specific to brain development.**

(A, B) Overall body weight (A) and brain weight (B) comparisons at P14, 24, and 30 between WT and NDUFS4 KO mice ($n = 5$ mice/group). (B) $P = 0.0327$ (P14), 0.0363 (P24), 0.1562 (P30)). (C) Half-hemisphere width mean measured between Bregma +1.53 and +1.23 mm ($n = 5$ mice/Bregma point/group). $P = 0.0134$ (P14), 0.0560 (P24), 0.0880 (P30), Row factor (0.0032). (D) Representative images from Bregma +1.23 mm of P14 mice. Scale bar = 500 μm. White line denotes where measurements across hemispheres were taken. In (A–C), data are presented as mean and standard error of the mean. *$P < 0.05$, ****$P < 0.0001$, ns not significant (two-way ANOVA). Source data are available online for this figure.

significant reduction in the proportion of NSPCs undergoing cell proliferation in NDUFS4 KO (Fig. 3C,D). Together, these findings suggest that complex I deficiency impairs NSPC production which, paired with our earlier findings, contributes to a delay/loss in newborn neurons during a critical postnatal period of brain development.

To corroborate our in vivo findings, we performed neurosphere assays to eliminate the influence of systemic and multicellular alterations on NSPC dynamics. Since our most striking findings were at P14, we focused all subsequent analyses on this age. NSCs were isolated from the SVZ of WT and NDUFS4 KO animals at 14 days of age and cultured in the same conditions to assess neurogenesis. Considering that NDUFS4 KO had fewer NSCs to begin with at this age, we plated and quantified the properties of secondary spheres after one passage to ensure we obtained the same starting number of cells prior to analysis. However, neurospheres derived from KO mice were still noticeably smaller and in lower numbers; quantification of sphere diameter from the spheres that were present conferred a significant decrease in size, illustrating a reduction in the capacity to self-renew (Fig. EV2A–C). When evaluating the distribution of

sphere size, there was a significant shift towards smaller spheres, which typically harbor progenitors that are nearing the limit of their self-renewal capacity (Mori et al, 2006; Pastrana et al, 2011). This may explain the inability of these cultures to expand beyond the first passage and further supports the notion that mitochondrial function is essential for postnatal production of NSPCs (Fig. EV2A). Taken together, the recapitulation of our in vivo proliferation findings (Fig. 3A,C) with the in vitro neurosphere assay suggests that disruption of NSPC production and lineage progression is governed in part by cell-intrinsic mechanisms.

## Single-cell transcriptomic profiling of postnatal SVZ identifies distinct classes and subclasses along the neuronal lineage affected in NDUFS4 KO mice

To understand how NDUFS4 impacts neurogenesis within the SVZ, which is occupied by numerous cell types with high transcriptional diversity in the postnatal brain, we performed single-cell transcriptomic profiling to better elucidate which biological processes

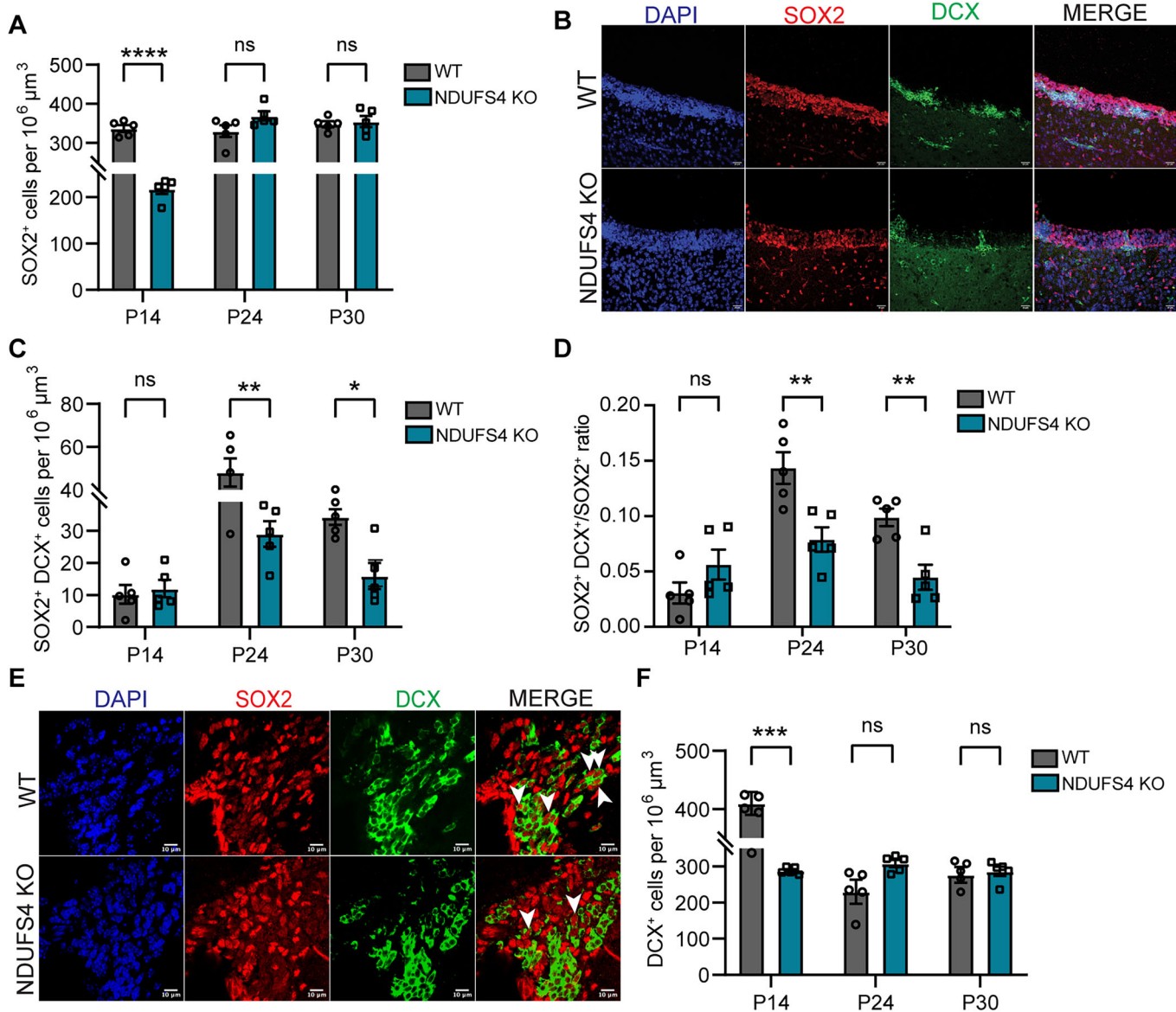

**Figure 2.   The SVZ displays a reduced number of neural progenitors, neuroblast density, and neural stem cell commitment in NDUFS4 KO mice.**

(A) Density of SOX2+ in the SVZ at P14, P24, and P30 ($n = 5$ mice/group). $P = < 0.0001$ (P14), 0.0761 (P24), 0.9694 (P30). (B) Representative confocal images of immunocytochemical detection of SOX2 (red), DCX (green), and DAPI (blue) of the lateral SVZ wall at P14 in WT and NDUFS4 KO mice. Scale bar = 20 μm. (C, D) Density of (C) double-positive cells (SOX2+/DCX+) and (D) double-positive (SOX2+/DCX+) /SOX2+ ratio at P14, P24, and P30 between groups ($n = 5$ mice/group). (C) $P = 0.9859$ (P14), 0.0080 (P24), 0.0112 (P30). (D) $P = 0.3341$ (P14), 0.0016 (P24), 0.0083 (P30). (E) Representative confocal images of immunocytochemical detection of SOX2 (red), DCX (green), and DAPI (blue) of the lateral SVZ wall at P24 between WT and NDUFS4 KO mice. White arrows indicate double-positive cells. Scale bar = 10 μm. (F) DCX+ cells in SVZ at P14, P24, and P30 ($n = 5$ mice/group). $P = 0.0006$ (P14), 0.0559 (P24), 0.9814 (P30). In (A, C, D, F), the bar plot represents the mean and standard error of the mean. Each dot represents one animal. *$P < 0.05$, **$P < 0.01$, ***$P < 0.001$, and ****$P < .0001$, ns not significant (two-way ANOVA). Source data are available online for this figure.

were most affected in its absence. To best align nomenclature, we use the terms neural progenitor cells (NPCs), transit-amplifying cells (TAPs), and Type C cells interchangeably. We collected 40,324 cells with a median of 1223 genes and 1842 unique molecular identifiers (UMIs) quantified per cell from WT and NDUFS4 KO from microdissected SVZ samples (Fig. 4A). For higher data resolution, we analyzed cells from all replicates, accounting for batch effects using Harmony. Initial analysis identified 19 distinct cell clusters (Fig. 4A), which were assigned to 14 known cell types

based on the detection of known marker genes (Yang et al, 2023) (Fig. 4A,B). We did not detect a significant difference between these clusters comparing genotypes when performing cluster proportion analysis (Fig. 4A).

Higher resolution of cell types within the neurogenic lineage was achieved by subclustering NSCs, TAPs, and NBs; resulting in nine neurogenic clusters (Basak et al, 2018; Llorens-Bobadilla et al, 2015), six of which represent different stem/progenitor states along the neurogenic lineage, such as quiescent vs active NSCs and early

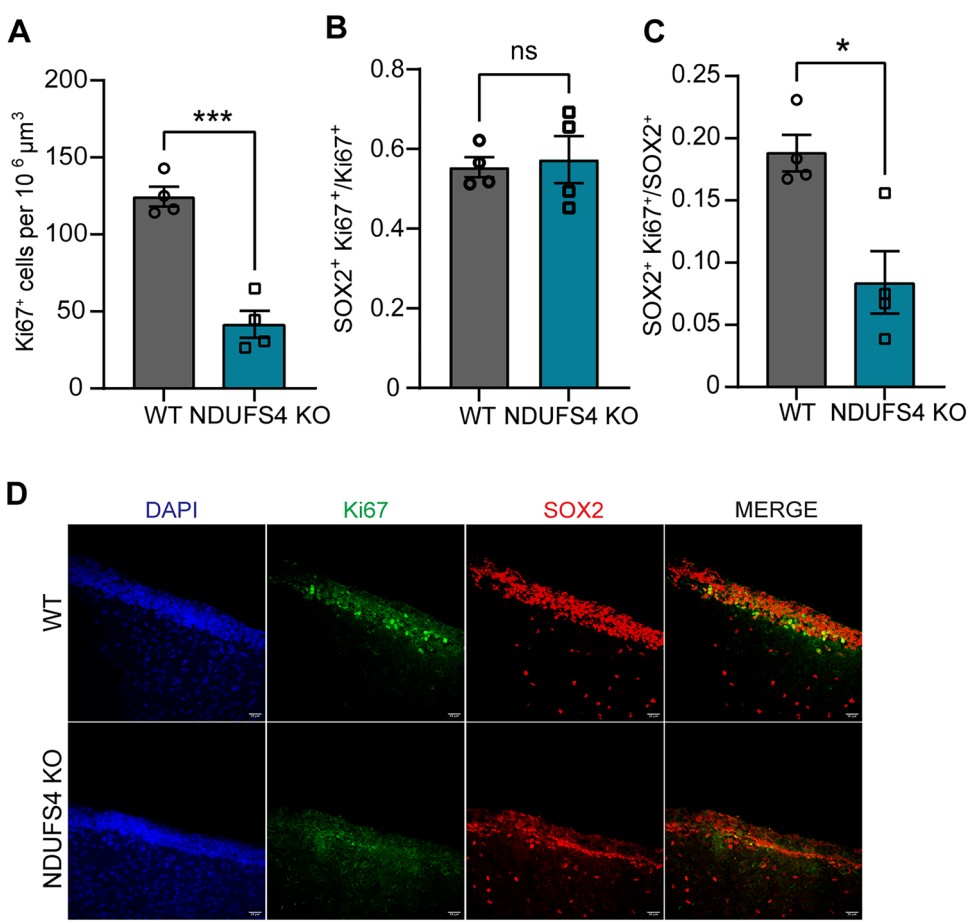

**Figure 3. Reduced Proliferation of Neural Progenitors in the SVZ of NDUFS4 KO mice.**

(A) Density of Ki67$^+$ cells in the P14 SVZ ($n = 4$ mice/group). $P = 0.0003$. (B) Percentage of SOX2$^+$ Ki67$^+$/Ki67$^+$ cells in P14 SVZ (C) SOX2$^+$ Ki67$^+$/ SOX2$^+$ ratio in the P14 SVZ ($n = 4$ mice/group). $P = 0.0118$. (D) Representative images from lateral ventricles of P14 mice stained for SOX2 (red), Ki67(green), and DAPI (blue). Scale bar = 20 μm. In (A–C), the bar plot represents the mean and the standard error of the mean. Each dot represents one animal. *$P < 0.05$, ***$P < 0.001$, ns not significant (unpaired $t$ test). Source data are available online for this figure.

vs late NBs (Fig. 4C). Subtypes were identified based on expression of (i) known genes (*Aqp4*, *Mki67*, and *Tubb3*) as well as (ii) genes associated with lipid biosynthesis, glycolysis, cell cycle, ribosome, mitosis, and neuronal differentiation (Fig. 4D,E). Further proportion analysis within the subclusters revealed a significant reduction in the percentage (33.88% vs 1.75%) of cells identified as early NBs paired with an increase in late NBs (2.40% vs 31.99%) in NDUFS4 KO compared with controls (Fig. 4F,G). Considering that late neuroblasts downregulate DCX, this data is in agreement with our histological data (Figs. 2F and EV1G). These data suggest a maturational bias in KO mice whereby late NBs represent most of the NB cell pool and agree with our decline seen in NSPCs undergoing the early lineage progression into NBs (SOX2$^+$ DCX$^+$) —presumably early NBs (Fig. 2F).

To further validate our findings suggesting proliferative impairments within the neuronal lineage, we compared the gene expression datasets of NSCs and TAPs for the cumulative fraction of genes associated with the G2/M border and M phases of the cell cycle. NDUFS4 KO NSCs and TAPs exhibited significantly lower cumulative expression of genes associated with these cell cycle phases (Appendix Fig. S1A,B). Paired with our previous evidence of hampered NSPC proliferation in vivo and in vitro, our findings suggest that NDUSF4 is a critical regulator of neurogenesis during early postnatal development that may be required for the production and maturation of key cell types within the neural lineage occupying the SVZ.

## Single-cell transcriptomic profiling demonstrates several dysregulated pathways along the neuronal lineage in NDUFS4 KO mice

To determine which signaling pathways may underlie our observed defects in neurogenesis, we analyzed differential gene expression profiles across the three main subclasses of cells impacted along the neuronal lineage in NDUFS4 KOs: NSCs, TAPs, NBs. We identified over 825 differentially expressed genes for NSCs and 613 genes for TAPs between genotypes (Fig. 5A–D; Datasets EV1 and EV2). These differentially expressed genes (DEGs) collectively rank neurogenesis as one of the top two most affected biological processes in NDUFS4 KO NSCs and TAPs, as shown by gene

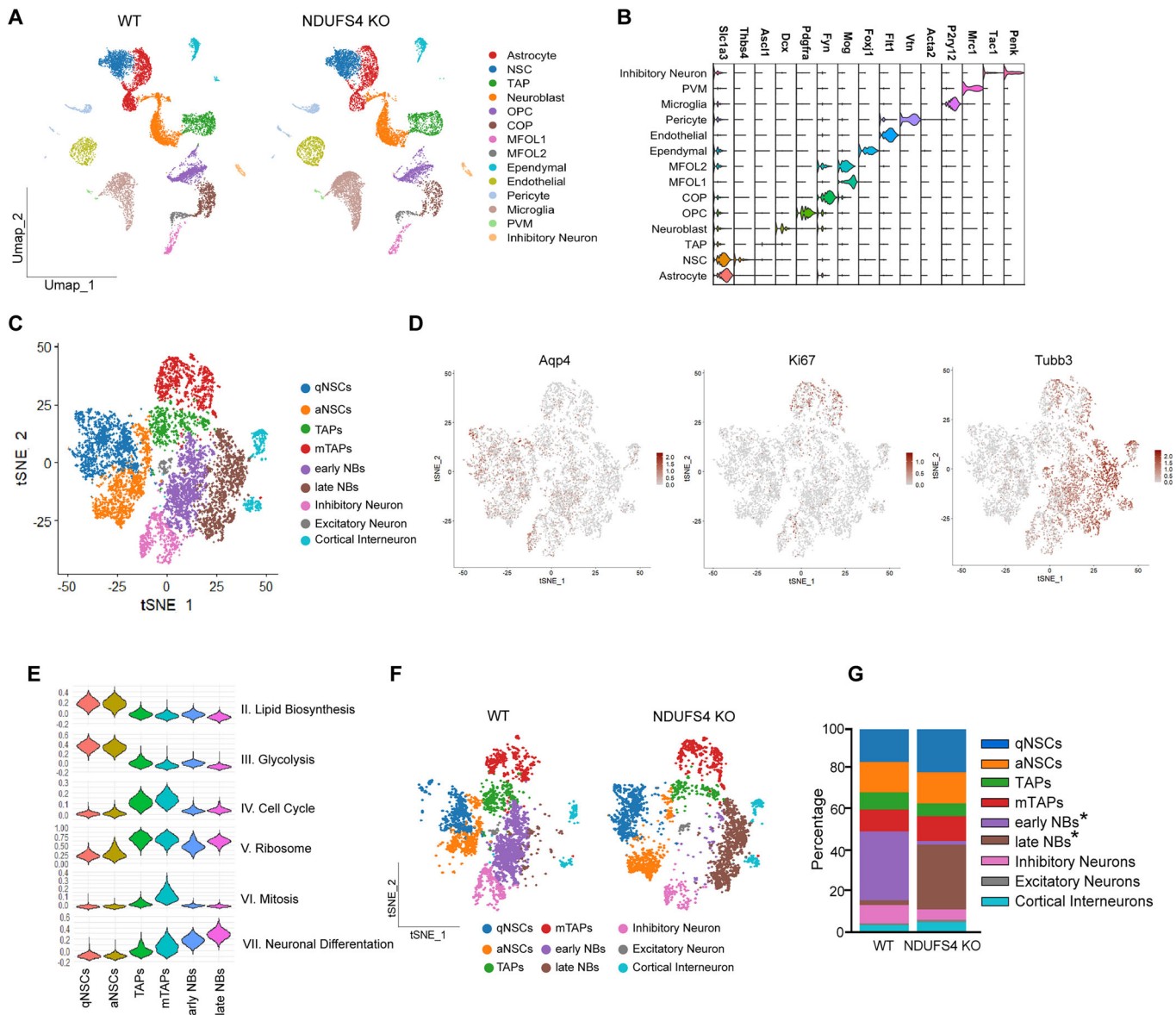

**Figure 4. Characterization of cell types and subtypes residing in the P14 SVZ following single-cell sequencing.**

(A) UMAP plots of WT and NDUFS4 KO cells colored by cluster annotation. NSCs neural stem cells, TAPs transit-amplifying progenitors, OPCs oligodendrocyte progenitor cells, COPs differentiation-committed oligodendrocyte precursors, MFOLs myelin-forming oligodendrocytes, PVMs perivascular macrophages. (B) Expression of known marker genes for each cluster. (C) tSNE plot for subclusters deriving from NSCs, TAPs, and NBs. (D) Expression of known markers of different subclusters. (E) Subcluster characterization based on published gene sets in (Llorens-Bobadilla et al, 2015) for relevant biological processes. (F) tSNE plots showing subtypes for WT and NDUFS4 KO separately. (G) Proportion of subtypes of NSPCs in WT and NDUFS4 KO. Source data are available online for this figure.

ontology (GO) analysis (Fig. 5B,D). Ingenuity pathway analysis (IPA) also revealed dysregulated genes in the KO NSCs associated with neurodevelopmental disorders; these genes have overlapping functions in neural cell proliferation, development, and differentiation (Fig. 5E,F).

Focusing on the neuronal lineage, we found a diverse range of pathways altered at different stages throughout immature neuronal production. For example, we observed upregulation of *Mt1* and *Mt3* in NSCs (Fig. 5A), which are genes previously reported to be highly expressed in radial glia precursors transitioning into quiescence during late embryonic stages (Yuzwa et al, 2017). An

upregulation in these genes suggests a possible premature shift toward quiescence in NDUFS4 KO NSCs. We also identified dysregulated upstream regulators of NSC survival (*Tcf7l2* and *Ctnnb1*, involved in Wnt signaling), stem cell self-replication (*Sox2*), proliferation (*Ar, Esr1, Npm1,* and *Mycn*), and quiescence (*Stk11*) (Chodelkova et al, 2018; Pagin et al, 2021; Zhuang et al, 2023) either as a DEG or predicted from IPA (Fig. 5B,E,F; Dataset EV1). Interestingly, DEGs in NSCs also revealed an increase in some subunits of Complex II-IV, which may indicate a compensatory mechanism (Calvaruso et al, 2012; van de Wal et al, 2025) in the absence of Complex I activity, where NDUFS4 is

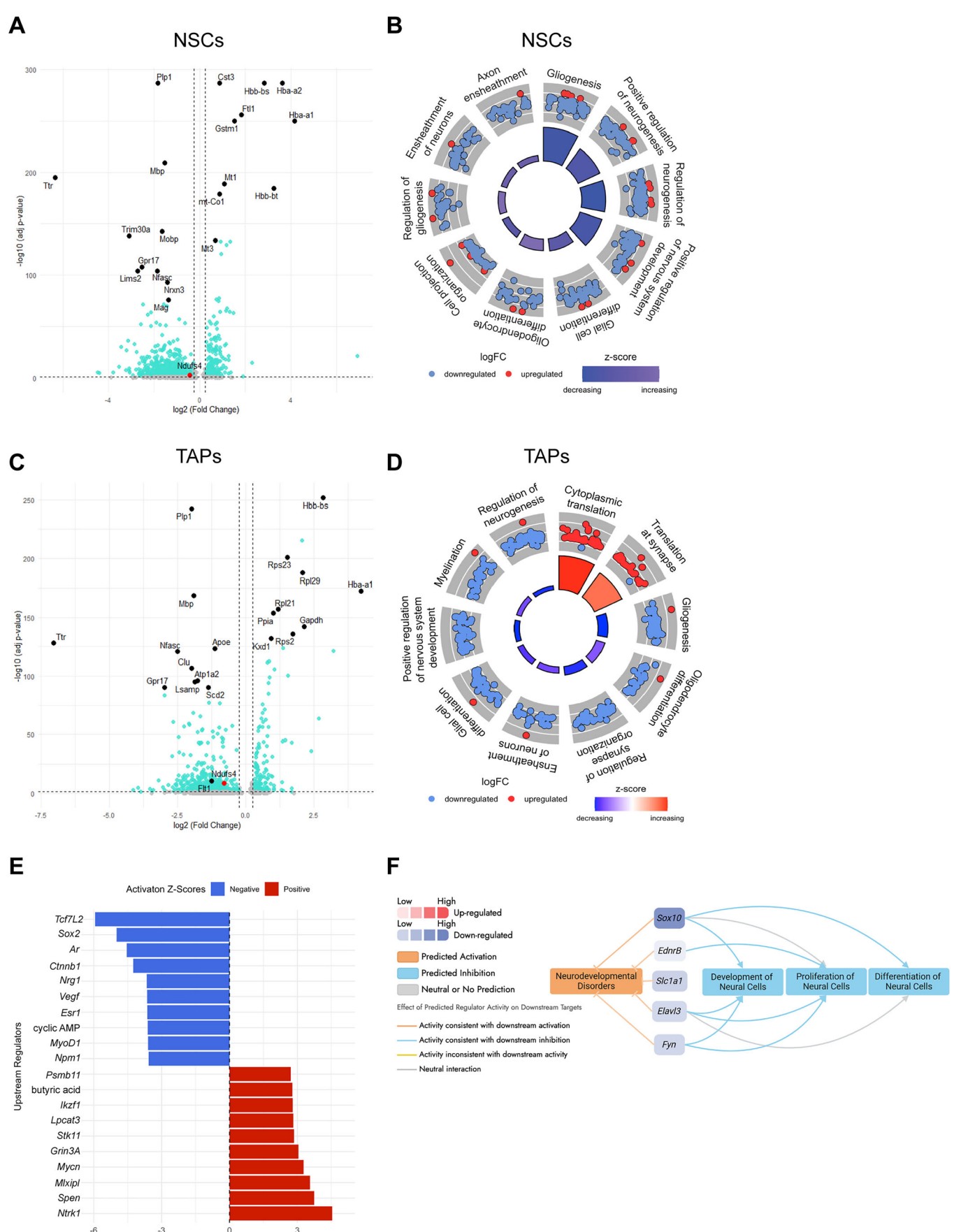

◀ **Figure 5. Differentially expressed genes and dysregulated biological pathways in NDUFS4 KO NSCs and TAPs.**

Volcano plot showing fold change of differentially expressed genes in NDUFS4 KO (**A**) NSCs and (**C**) TAPs. ($n = 1$ per group, then matched against the other replicate. Ones that are in the same direction were selected). Top 10 up- and downregulated genes are labeled (black circles) and *Ndufs4* gene is shown in red circle. The thresholds were set at log2(Fold Change) ≥ +0.25 and ≤ -0.25, and −log10 (adj *P* value) <1.2991. GO circle plot showing top dysregulated biological processes and regulation of associated genes in NDUFS4 KO (**B**) NSCs and (**D**) TAPs. (**E**) Top 10 up- and downregulated upstream regulators in KO NSCs from ingenuity pathway analysis. (**F**) Dysregulated genes in KO NSCs are associated with predicted neurodevelopmental disorders that overlap with neural cell proliferation, development, and differentiation. In (**A, C**), the Wilcoxon rank-sum test (nonparametric) was performed. Source data are available online for this figure.

significantly decreased (Fig. 5A,C; Appendix Fig. S2). *Cst3* is another upregulated gene in NSCs important for brain patterning and neuronal maturation, which is positively correlated with neurodevelopmental delays in preterm males (Kato et al, 2006; Zhu et al, 2023). Additionally, we observed downregulation of *Ttr* in both NSCs and TAPs, which has been implicated in neural stem cell fate decisions (Vancamp et al, 2019). Other top downregulated genes for TAPs indicate an impaired reprogramming to oligodendrocytes, such as (*Cdk18* and *Bcas1*) and axon to myelin communication (*Gjc3*) (Altevogt et al, 2002; Fard et al, 2017; Pan et al, 2019) (Fig. 5C,D; Dataset EV2). One top upregulated gene *Lin28* participates in Wnt signaling regulation, which when excessively upregulated causes lissencephaly and abnormal neurodevelopment (Bhattacharya et al, 2018; Greco et al, 1996; Lee et al, 2019) (Fig. 5C,D; Dataset EV2). The KO NBs displayed increased expression of 388 genes, mostly associated with increased translation, which may account for the observed push towards late-stage neuroblasts (Baser et al, 2019) (Fig. EV3A,B; Dataset EV3).

Owing to the diverse classes and subclasses of cells identified, we next wanted to evaluate potential alterations in intercellular interactions. A disruption in cell-to-cell communication was inferred between ligand and receptors upon further bioinformatic analyses between all identified cell types. There was an overall reduction in interactions among cells of neurogenic lineages, including reduced crosstalk among NSCs, TAPs, neuroblasts, and neurons (Fig. EV4A,B). This reduced interaction led to altered information flow in signaling pathways involved in the regulation of NSPCs. For example, we observed an upregulation of signaling associated with neural cell adhesion molecules (NCAMs) (Fig. EV4C), which has been implicated in decreased progenitor proliferation in the neurogenic stem cell niche (Amoureux et al, 2000). However, there was an increased interaction coming from inhibitory neurons to neuroblasts, potentially influencing dendrite growth and survival of newborn neurons. Outside of the neurogenic lineage, there was increased signaling between microglia and committed oligodendrocyte progenitors (COPs) as well as microglia and neurons (Fig. EV4A,B). This rewiring is mirrored by a KO-skewed rise in inflammatory ligand–receptor programs (e.g., TNF, CXCL, CCL, TGFβ), which could potentially dampen progenitor proliferation, migration, and survival (Fig. EV4C). There was also an increase in signaling associated with vascular and endothelial activation (VCAM, PECAM1, ICAM) which can influence neural and glial progenitors by increased docking (Bjornsson et al, 2015; Kokovay et al, 2012) (Fig. EV4C). In addition, we observed an increased crosstalk involving differentiation of COPs and neuroblasts, both of which are destined to migrate out of the SVZ niche and contribute to white matter remodeling (Fig. EV4A). These analyses help contextualize the cellular milieu within the SVZ in

NDUFS4 KO animals, whereby loss of complex I activity in mitochondria has not been restricted to a single cell type; therefore, it is also important to consider potential confounds from extrinsic signaling mechanisms. Together, these findings suggest that the postnatal neurogenic niche is changed in composition, altering the genetic profile of the stem cell pools, thus resulting in a completely different neurodevelopmental signaling signature in LS than in neurotypical development. We next focused our attention on the second most affected biological process in NDUFS4 KO NSCs: gliogenesis.

## Decreased oligodendrogenesis in NDUFS4 KO mice

Because (i) neurons and oligodendrocytes (OLs) are both derived from NSPCs, (ii) our bioinformatic assessments indicated a significant downregulation in gliosis within the NSC and TAP populations, and (iii) we found a significant reduction in NSPC numbers throughout the SVZ (Figs. 2A,F and EV1A,G), we reasoned that oligodendrogenesis and functional maturation into myelin-producing OLs would be effectively downregulated in the absence of Complex I activity. Recently, remodeling of mitochondrial content, dynamics, and organization has been reported during oligodendrocyte development in vivo (Bame and Hill, 2024). Considering that OLs increase in surface area by up to 6500-fold during maturation/myelination (Chong et al, 2012), additional insight into the role mitochondria play during these processes is warranted.

We first performed immunohistochemistry to assess the total number of OLs throughout the entire OL lineage within the SVZ. Quantification of OLIG2$^+$ cells exhibited no significant difference in the density of OLs between WT and NDUFS4 KOs (Fig. EV5A,C). We next assessed the oligodendrocyte progenitor (OPCs) population and found no differences in PDGFRα$^+$ OPCs (Fig. EV5B,C). While OPCs are generated within the SVZ, they must migrate to target axonal tracts to commence maturation and functional myelination; in addition, OPCs are capable of self-renewing following emigration from their points of origin (Morrison et al, 2020). In considering the dynamic cell classes and transcriptional profiles generated when clustering for neuronal lineage (Fig. 4C), we reasoned that there may be an imbalance in specific subclasses of OLs that would not be accounted for by total cell population analyses. Therefore, we utilized a combination of birth dating transcription factors to assess the embryonic origins and predictive cell fates of stem/progenitor cells within the SVZ.

While oligodendrogenesis commences embryonically (~E12.5), myelination of the mouse brain occurs postnatally and is critical for proper motor and cognitive functions (Kessaris et al, 2006). There are three known waves of oligodendrocytes derived from distinct NSPC populations at different stages of perinatal brain

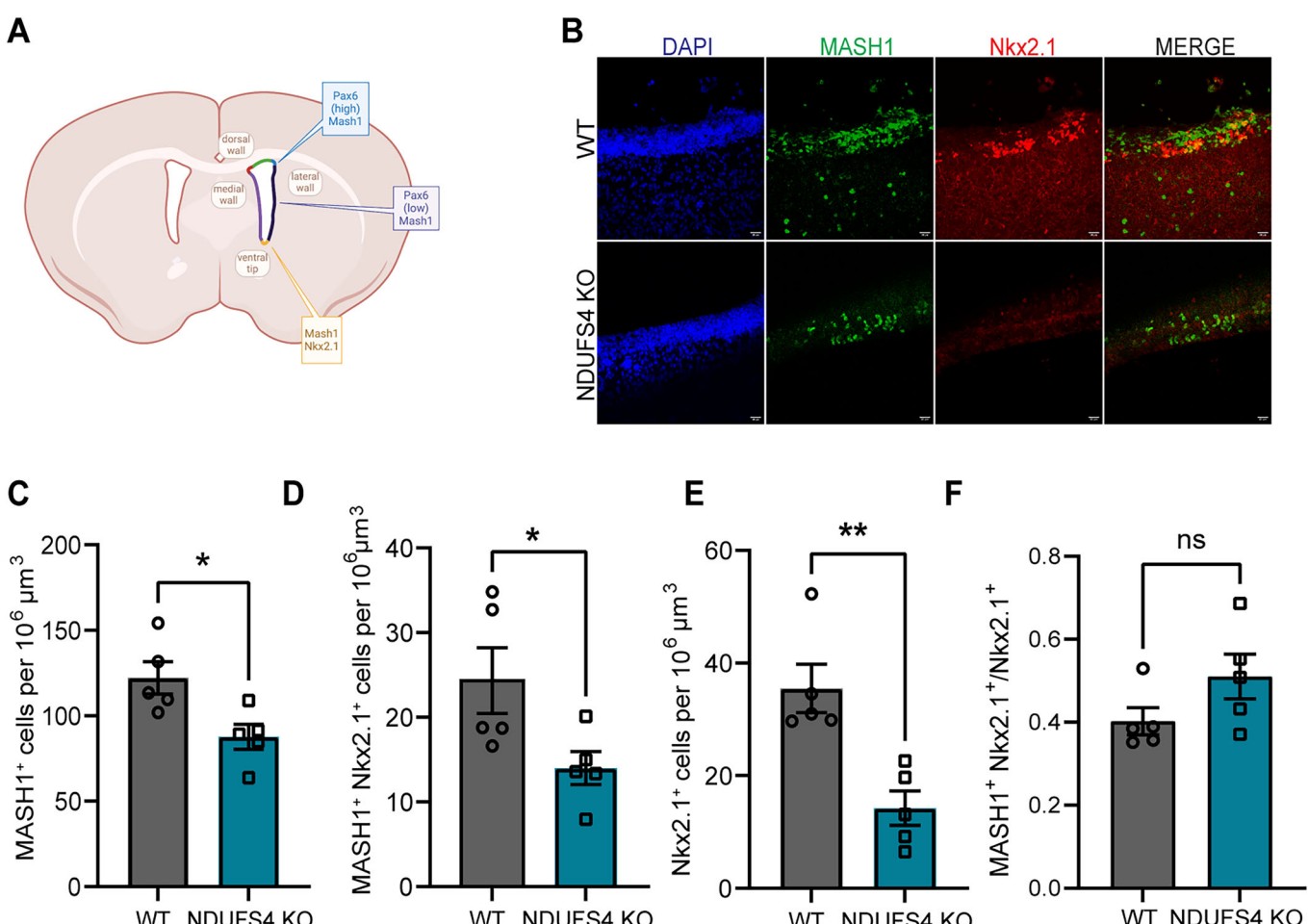

**Figure 6. NSCs deriving from the embryonic lateral and medial ganglionic eminences were reduced in the postnatal SVZ of NDUFS4 KO mice.**

(A) Schematic illustrating the expression patterns of MASH1, Pax6, and NKX2.1 in the lateral, ventral, and dorsal walls of the SVZ. (B) Representative confocal images of immunohistochemical staining for MASH1 (green), Nkx2.1 (red), and DAPI (blue) in the lateral SVZ of WT and NDUFS4 KO mice at P14. Scale bar = 20 μm. (C–E) Quantification of cell densities in the SVZ at P14: (C) MASH1+ cells, (D) double-positive Nkx2.1+ MASH1+ cells, (E) Nkx2.1+ cells. (F) Ratio of double-positive Nkx2.1+ MASH1+ cells to Nkx2.1+ cells ($n = 5$ mice/group). (C) $P = 0.0196$. (D) $P = 0.0448$. (E) $P = 0.0038$. In (C–F), bar plot represents mean and standard error of mean. Each dot represents one animal. In (C–E), unpaired $t$ test was performed. In (F), Mann–Whitney $U$ test was performed. *$P < 0.05$, **$P < 0.01$, ns not significant. Source data are available online for this figure.

development; in addition, these waves immigrate from different regional sources, including the lateral ganglionic eminence (LGE), medial ganglionic eminence (MGE), and the postnatal cortex (Kessaris et al, 2006). Importantly, NSCs in the postnatal brain are in part derived from embryonic LGE/MGE, maintain their spatial organization (LGE, lateral wall, MGE, ventral tip) (Fig. 6A), and express some of the same markers (Stenman et al, 2003; Waclaw et al, 2009). Therefore, we performed IHC targeting the transcription factors MASH1 (LGE and MGE) and Nkx2.1 (MGE), at P14, a time in development when the corpus callosum is heavily populated with MGE-derived (~50%) cells of the OL lineage.

We found a significant reduction in Nkx2.1+, MASH1+, and MASH1+ Nkx2.1+ cells, indicating that NSCs derived specifically from the LGE (MASH1+ Nkx2.1+) and MGE (Nkx2.1+) were affected in the NDUFS4 KO SVZ (Fig. 6B–E). We found no significant differences in the ratio of MASH1+ Nkx2.1+ to Nkx2.1+ (40.23% vs 50.01%), suggesting that these populations were seeded

in the SVZ in a similar manner from each embryonic source assessed (e.g., - LGE, MGE), with equivalent proportions of OL committed cells within the NSC pools (Fig. 6F). These findings are within anatomically equivalent planes of the LGE and MGE - the predecessor tissues to the SVZ—around a postnatal age in which the source of OLs populating the corpus callosum (CC)—the nearest axonal tract to be myelinated—are shifting from primarily LGE-derived to MGE- and cortically-derived (Kessaris et al, 2006). MASH1 is an essential promotor of oligodendrogenesis and subsequent myelination (Nakatani et al, 2013), and MASH1+ progenitors contribute to both neurons and oligodendrocytes postnatally (Parras et al, 2004); therefore, a loss in the number of MASH1+ cells may have an impact on both oligodendrogenesis and neurogenesis, defining a cell type that lies at the intersection of the mechanism by which Complex I deficiency results in brain maturational delays evident prior to the onset of the neurodegenerative signatures in LS.

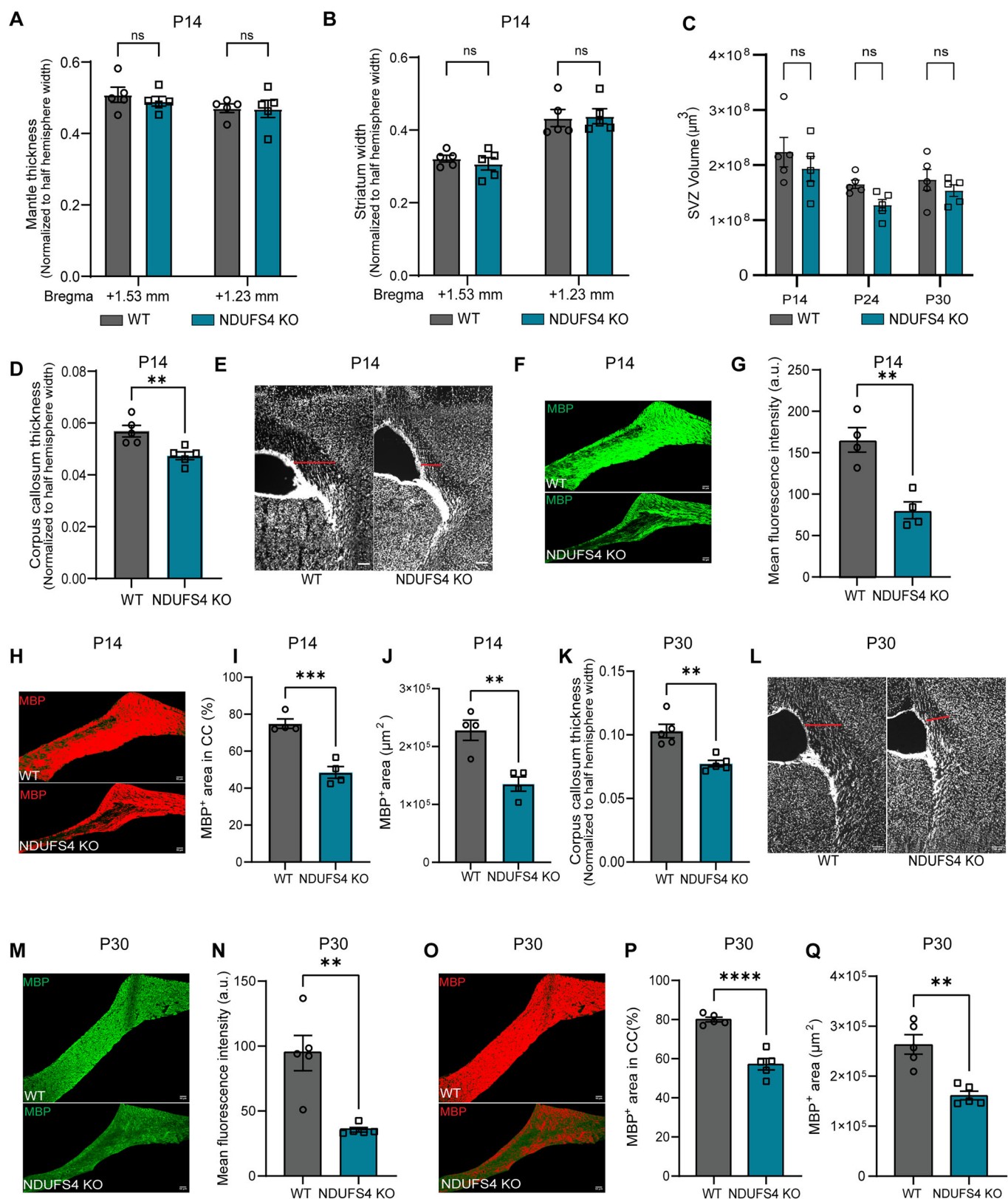

**Figure 7. Reduced Myelination in NDUFS4 KO corpus callosum.**

(A–D) Measurements of neuroanatomical structures from coronal sections at +1.53 mm and +1.23 mm from bregma: (A) Mantle thickness, (B) Striatum width. (C) Estimated SVZ volume measured across five coronal sections per animal. Two-way ANOVA was performed; ns not significant. (D) Corpus callosum thickness measured from P14 brain coronal sections at +1.23 mm from bregma ($n = 5$ mice/group). $P = 0.0078$. (E) Representative confocal images of the corpus callosum from WT and NDUFS4 KO mice at P14. Scale bar = 100 μm. The red dotted line represents the measurement taken for this image. (F) Representative confocal images of immunohistochemical staining for MBP in the corpus callosum at P14. Scale bar = 50 μm. (G) Quantification of mean fluorescence intensity of MBP in corpus callosum at P14 ($n = 4$ mice/group). $P = 0.0032$. (H) Threshold optimized confocal images of WT and NDUFS4 KO corpus callosum presenting area coverage by MBP fluorescence at P14. Scale bar = 50 μm. (I) Percentage of area covered by MBP fluorescence in corpus callosum at P14 ($n = 4$ mice/group). $P = 0.0007$. (J) Mean area coverage of MBP in CC at P14 ($n = 4$ mice/group). $P = 0.0049$. (K) Corpus callosum thickness measured from P30 brain coronal sections at +1.23 mm from bregma ($n = 5$ mice/group). $P = 0.0025$. (L) Representative confocal images of the corpus callosum from WT and NDUFS4 KO mice at P30. Scale bar = 100 μm. The red dotted line represents the measurement taken for this image. (M) Representative confocal images of immunohistochemical staining for MBP in the corpus callosum at P30. Scale bar = 50 μm. (N) Quantification of mean fluorescence intensity of MBP in corpus callosum at P30 ($n = 5$ mice/group). $P = 0.0026$. (O) Threshold optimized confocal images of WT and NDUFS4 KO corpus callosum presenting area coverage by MBP fluorescence at P30. Scale bar = 50 μm. (P) Percentage of area covered by MBP fluorescence in corpus callosum at P30 ($n = 5$ mice/group). $P = < 0.0001$. (Q) Mean area coverage of MBP in CC at P30 ($n = 5$ mice/group). $P = 0.0013$. In (A–C), the bar plot represents the mean and the standard error of the mean. Each dot represents one animal. ns not significant (two-way ANOVA). In (D, G, I–K, N, P, Q), the bar plot represents the mean and standard error of the mean. Each dot represents one animal. *$P < 0.05$, **$P < 0.01$, ***$P < 0.001$, and ****$P < .0001$ (unpaired $t$ test). Source data are available online for this figure.

## Complex I deficiency results in disrupted myelination of the corpus callosum

Because MGE-derived OPCs migrate to and myelinate the corpus callosum, and we found a significant reduction in *Mbp* gene expression—one of the most abundant myelin proteins in the brain—in our single-cell seq data (Fig. 5A), we next assessed the CC. Corpus callosum formation begins prenatall,y whereby the initial axons cross the brain midline and continue through early postnatal days, with migrating OPCs from the dorsal wall populating this axonal tract and generating mature myelinating oligodendrocytes (Hoshino et al, 2024). We reasoned that an impairment in either of these processes—neurons sending axons through the CC or loss of myelin—would result in an overt anatomical phenotype.

Considering our findings of reduced brain hemispheric width in KO animals (Fig. 1C, D), we measured landmark neuroanatomical structures on the coronal planes corresponding with the Bregma areas where we found the most significant differences (+1.53 and +1.23) (Fig. EV1A). We found no differences in cortical mantle thickness, striatal width, or SVZ volume (Fig. 7A–C). When assessing the CC, we found a significant reduction in thickness in NDUFS4 KO compared to WT at P14 (Fig. 7D,E). We next performed immunostains for MBP to visualize myelin protein at P14 (Fig. 7F). Quantification of mean fluorescence intensity (Fig. 7G), percentage area (Fig. 7H,I)—to account for differences in CC size—covered, and mean area coverage (Fig. 7J) revealed a significant reduction in MBP expression and quantity, indicating a disruption in myelination (Fig. 7G–J). The deficit observed in the CC at P14 persisted through P30 (Fig. 7K, L). This reduction was accompanied by a corresponding decrease in MBP expression and diminished MBP coverage across the CC area (Fig. 7M–Q). Together, these findings indicate that Complex I activity is critical for OL production and myelination within the postnatal SVZ and CC. While agenesis of the CC has been documented in a few LS cases, the areas of the brain most assessed and affected are deep brain structures seeded with neurons early in embryonic development (Kakkar et al, 2022). While myelination is strictly a postnatal phenomenon in the mouse brain, OL lineage progression follows a

series of steps akin to neurons which vary in energetic demand at each stage. Therefore, diffuse white matter injury may represent a largely overlooked phenotype in monitoring human neonatal brain development and a defect that precedes the pediatric neurodegenerative manifestations most ascribed to motor and cognitive impairments in LS.

## Discussion

Using the preclinical mouse model of LS, we investigated the neurodevelopmental underpinnings of the disease postnatally, considering that new reports indicate patients do not solely display degenerative regressive symptomology. Our work demonstrates that prior to the appearance of neurodegenerative phenotypes, neural stem and progenitor function and differentiation is perturbed postnatally and the temporal timing of typical neurodevelopmental events are delayed during LS. Our data indicate that the disruption of NDUFS4 disrupts cell type composition of neurogenic niches as well as the transcriptomic signatures of the NSPCs, altering the phenotypes of these cells.

Our data agree with others that have reported the sensitivity of NSPCs and stem cells to mitochondrial dysfunction, albeit those models are more severe and unphysiological. In models with double-strand break damage to mitochondrial DNA (mtDNA), highly proliferative stem cell populations appeared most affected (Pinto et al, 2017; Wang et al, 2013). Neural and hemopoietic stem cell defects were described in the Mutator mice, where a proofreading-defective polymerase γ causes a high level of mtDNA mutations (Ahlqvist et al, 2012; Norddahl et al, 2011). Conditional ablation under the control of NSPC promoters of AIF, mitochondrial fission or fusion genes, such as *Mfn1/Mfn2* and *Drp1*, or pharmacological disruption of fission and fusion halt or perturb the differentiation process and ability for self-renewal (Iwata et al, 2020; Khacho et al, 2016). All these previous studies highlighted the importance of mitochondria in stem cell biology, but our study highlights that in diseases and disorders where mitochondrial dysfunction affects post-mitotic tissues, alterations in stem cells should not be overlooked.

While lesions within the gray matter are evident in LS patients and recapitulated in the mouse model, less attention has been given to the white matter essential for neurotypical motor and cognitive development. With consideration of how neuronal and oligodendroglial developmental programs unfold in sequence in mice, our collective findings - reduced Mash1$^+$ OLs in the SVZ, no changes in SVZ volume, and no changes in total OL population within the SVZ - indicate a disruption (i) in local oligogenesis within the CC, (ii) stall in lineage progression, (iii) inability to generate myelin, (iv) loss of myelin initiating cues from unhealthy/dysmature axons or a combination of cellular stress points. Further conditional genetic studies paired with axonal tracers will greatly aid in deciphering the full impact of the eclectic changes in transcriptomic profiles identified. While it is known that each of the 3 competitive waves of OPC development can functionally replace one another (Kessaris et al, 2006), which may explain why we only see differences in a proportion of OPCs rather than whole population (PDGFRα), our MBP findings suggest that this phenotype is more representative of a failure to mature or myelinate the CC subsequent from migration from SVZ.

Dysmyelination of the CC may be dampened in humans with LS compared to this mouse model as humans are born with a mixed profile of OLs including progenitors, as myelination commences during the 2$^{nd}$ trimester (Grotheer et al, 2023; Yarnykh et al, 2018); therefore, the stress imposed by their *ex utero* environment may manifest in delays in myelination rather than complete lack of corpus callosum formation. Recent evidence demonstrates that loss of the *Nfia1* gene results in agenesis of the corpus callosum (das Neves et al, 1999), and our transcriptomics demonstrate a significant reduction in this gene within the TAPs, suggesting that this progenitor pool, capable of producing oligodendrocytes and neurons, may react to Complex I deficiency differently before and after birth.

However, we cannot rule out the possibility that demyelination is occurring in this model, which would require time points prior to P14 and electron microscopy to verify or rule out. But neuroinflammatory mechanisms that facilitate demyelination in the region of the brain we are studying are not present at this time point (Aguilar et al, 2022; Quintana et al, 2010). Adaptive immunity is not a major player in the pathogenesis of the NDUFS4 KO mice nor is the blood brain barrier breached, which would all be expected if demyelination accounted for WM defects described here (Hanaford et al, 2025; Reynaud-Dulaurier et al, 2024). In addition, we cannot rule out the possibility that early callosal neuronal stress may induce demyelination of their respective axons.

Postnatal myelination of the corpus callosum commences near P11 in mice and is regulated in part by OPC-encoded hypoxia inducible factor (HIF), illustrating a significant role for appropriate oxygen levels during neurotypical white matter development (Sturrock, 1980; Yuen et al, 2014). In addition, OPCs play a role in regulating angiogenesis within the corpus callosum, attracting new blood vessels to deliver additional oxygen to their milieu, which is governed by OPC-specific HIF1/2α (Yuen et al, 2014). These studies utilized chronic hypoxic rearing from P3 to P11 and are reminiscent of hypoxic breathing therapy in NDUFS4 KO mice. Previous studies using hypoxia therapies in NDUFS4 KOs utilize mice at P30 to preclude any confounding factors related to hypoxia-induced vascular alterations documented in early development (Ferrari et al, 2017; Jain et al,

2016). Combined with our findings, we speculate that in addition to disrupted myelination, there may be a reduction in vascularization of the corpus callosum; such a finding would be intriguing as angiograms with contrast agents focused on the corpus callosum are clinically feasible/tractable in young LS patients and may offer additional insights into the presymptomatic stage of this disease.

Recent genetic crossings with NDUFS4 KO mice to disrupt PHD oxygen-sensing enzymes demonstrated that hypoxic breathing therapy to overcome brain hyperoxia in these mice is effective independent of HIF activation (Jain et al, 2019). In addition, carbon monoxide as well as anemia reversed the disease as an alternative strategy to reduce oxygen delivery to the brain (Jain et al, 2019). Such foundational studies and others have paved the way for leading theories involving oxygen tension and brain hyperoxia in LS. The potential inability to offload circulating oxygen bound to hemoglobin in the case of Complex I deficiency resulting in parenchymal hyperoxia and ultimately cytotoxicity owing to ROS accumulation merits further exploration. While there is strong evidence for this phenomenon in older ≥P30 NDUFS4 KO mice, this remains unexplored near birth when mammals transition from fetal hemoglobin to normal pulmonary ventilation following the first breath of *ex utero* life. Therefore, future studies focused on the potential role of increased oxygen tension following natural birth may identify the very onset of $O_2$ overaccumulation in the brain, where mitochondria are incapable of proper $O_2$ consumption. In addition, probe-based therapies such as HypoxyStat to uncouple the functional roles played by HIFs will offer new strategies to bypass the impractical utility of hypoxia breathing therapy in humans along with the unwanted systemic alterations, as well as provide novel insights in comparison to hyperoxia models, whereby the lungs may also be damaged (Blume et al, 2025).

The CC is a relatively small structure to quantify by standard clinical MRI assessments in the young and, unlike more noticeable cystic white matter lesions, may require imaging modalities such as diffusion tensor imaging (DTI) to accurately assess diffuse changes in myelination/integrity. Such prospective clinical studies in LS patients in parallel with well-designed/characterized genetic murine models of mitochondrial dysfunction in the young will be invaluable in developing preventative and predictive treatment strategies to improve psychomotor outcomes.

## Limitations of the study

We found that postnatal neurodevelopmental processes were disrupted in a preclinical LS mouse model. More thorough examination embryonically and at the other postnatal neurogenic niche in rodents, like the hippocampus, is warranted. Another study knocking out NDUFS2 under the control of the overexpressed transgenic human *GFAP* promoter described neurogenic postnatal defects, but the model produced a severe impact embryonically, starting at day 13.5 (Cabello-Rivera et al, 2019; Zhuo et al, 2001). It is also unclear if the severity of the defects described in this study extends to LS caused by other mutations that are not Complex I specific. However, we highly find that this is likely due to recent work describing NSPC defects in an iPSC organoid model harboring *SURF1* mutations (Inak et al, 2021).

# Methods

## Reagents and tools table

| Reagent/resource | Reference or source | Identifier or catalog number |
|---|---|---|
| **Experimental models** | | |
| Mouse wild-type and homozygous for Ndufs4 from Ndufs4 +/− matings | Jackson Laboratory | B6.129S4-Ndufs4tm1.1Rpa/J |
| **Recombinant DNA** | | |
| N/A | | |
| **Antibodies** | | |
| Rabbit anti-Sox2 | Millipore Sigma | AB5603 |
| Guinea pig anti-DCX | Millipore Sigma | AB2253MI |
| Mouse anti-MASH1 | BD Biosciences | 556604 |
| Rabbit anti-Nkx2.1 | Fisher Scientific | 7601MI |
| Mouse anti-Olig2 | EMD Millipore | AB9610 |
| Rat anti-PDGFRα | BD Pharmigen | 558774 |
| Mouse anti-Ki67 | BD Pharmigen | 550609 |
| Rabbit anti-myelin basic protein | Cell Signaling Technology | 78896 |
| **Oligonucleotides and other sequence-based reagents** | | |
| N/A | | |
| **Chemicals, enzymes, and other reagents** | | |
| HBSS without $Ca^{2+}$ and $Mg^{2+}$ | Fisher Scientific | SH3058801 |
| Neural Tissue Dissociation Kit (P) | Miltenyi Biotec | 130-092-628 |
| Beta-mercaptoethanol | Fisher Scientific | AC125472500 |
| Bovine serum albumin | Fisher Scientific | AM2616 |
| HBSS containing $Ca^{2+}$ and $Mg^{2+}$ | Thermo Scientific | 14025092 |
| HEPES | Fisher Scientific | 15630080 |
| Percoll | Fisher Scientific | 45001754 |
| PIPseq T20 3′ Single Cell RNA Kit v4.0PLUS, 4 rxn | PIPseq | FBS-SCR-T20-4-V4.05 |
| RNase inhibitor | Millipore Sigma | 3335399001 |
| DMEM/F12 | Gibco | SH3026101 |
| N2 supplement | Fisher Scientific | 17502048 |
| EGF | R&D Systems | 236-EG-200 |
| FGF | R&D Systems | 233-FB-025/CF |
| Accutase | STEMCELL Technologies | 07920 |
| OCT media | Tissue-Tek | 25608-930 |
| Tyramide Signal Amplification kit | ThermoFisher Scientific | B40932 |
| Qubit 1× dsDNA High Sensitivity Assay Kit | ThermoFisher Scientific | Q33230 |
| BioAnalyzer High Sensitivity DNA Kit | Agilent | 5067-4626 |
| DAPI Fluoromount-G | SouthernBiotech | 0100-20 |
| **Software** | | |
| PIPseeker™ v3.3 | PIPseq | |

| Reagent/resource | Reference or source | Identifier or catalog number |
|---|---|---|
| STAR aligner | https://github.com/alexdobin/STAR | |
| Seurat (v1.4) | Satija lab | |
| edgeR | Bioconductor | |
| scCODA v0.1.9 | Theis lab | |
| CellChat | Nie lab | |
| Ingenuity Pathway Analysis | Qiagen | |
| **Other** | | |
| Ultra-low adherent 24-well culture plate | Fisher Scientific | NC1882826 |

## Lead contacts and materials availability

Further information and requests for resources and reagents should be directed to and will be fulfilled by Lead Contacts, Paul D. Morton, Ph.D. (pmorton@vt.edu) and Alicia M. Pickrell, Ph.D. (alicia.pickrell@vt.edu).

### Animals

Wild-type (WT) and heterozygous (Het) mice for the *Ndufs4* gene (B6.129S4-Ndufs4tm1.1Rpa/J) were obtained from Jackson Laboratory. Mice were housed under standard conditions, maintained on a 12-h light/dark cycle, with ad libitum access to food and water. To generate NDUFS4 knockout (KO) animals, heterozygous *Ndufs4* mice were crossed. For immunohistochemical analysis, mixed sex was used. Single-cell sequencing samples were collected from the following sexes: I) WT1 was derived from one male and one female WT mouse, II) WT2 was derived from a single female WT mouse, III) NKO1 and NKO2 were derived from two different female NDUFS4 KO mice. All experimental procedures were conducted in accordance with the NIH Guide for the Care and Use of Laboratory Animals and were approved by the Virginia Tech Institutional Animal Care and Use Committee under protocol 23-111.

### Immunohistochemistry

To prepare brain sections for IHC, mice were anesthetized with a ketamine/xylazine cocktail and perfused with cold 1× PBS containing 1000 IU/mL heparin. The brains were harvested and incubated in 4% paraformaldehyde (PFA) overnight at 4 °C. After fixation, the brains were incubated with 30% sucrose in PBS for 24 h until the brain sunk and then frozen in OCT media (Tissue-Tek) containing 30% sucrose. Frozen brains were sectioned and mounted at a thickness of 30 μm and stored at −80 °C until further use. For antigen retrieval, sections were incubated in 95 °C sodium citrate buffer (pH 6) for 5 min. Tissue sections were permeabilized with 0.4% Triton X for 10 min followed by incubation in blocking buffer (5% Bovine serum albumin + 2.5% Normal goat serum) for 1 h. Tissue sections were incubated with primary antibodies at the following dilutions: rabbit anti-Sox2 (1:250; Millipore Sigma #AB5603), guinea pig anti-DCX (1:350; Millipore Sigma #AB2253MI), mouse anti-MASH1 (1:200; BD Biosciences #556604), rabbit anti-Nkx2.1 (1:300; Fisher Scientific #07601MI),

mouse anti-Olig2 (1:250, EMD millipore #AB9610), rat anti-PDGFRα (1:200; BD Pharmigen #558774), mouse anti-Ki67 (1:250; BD Pharmigen #550609), and rabbit anti-Myelin Basic Protein (1:50, Cell signaling Technology #78896). All secondary antibodies were used at a 1:200 dilution. A standard tyramide signal amplification (TSA) protocol (ThermoFisher Scientific) was employed as per the manufacturer's instructions whenever primary antibodies from the same host species were used. Stained tissues were mounted using DAPI Fluoromount-G (SouthernBiotech #0100-20).

### Imaging

Following immunostaining, high-resolution Z-stack images (1 μm step size) of the dorsolateral, lateral, and dorsal (for PDGFRα and Olig2 staining) walls of the subventricular zones (SVZs) were captured using a Nikon C2 confocal laser scanning microscope (Nikon Instruments, Melville, NY) at ×40 magnification. Reference maps at ×10 magnification were used to guide the acquisition of the ×40 images. Neurospheres were imaged under a bright-field filter using a ×10 objective.

The Nikon C2 is outfitted with bright-field and phase contrast with a Kinetix CMOS camera and fluorescent channels DAPI, GFP, Texas Red, and Far Red. The Nikon LUN4 has a four-line solid-state laser system with Perfect Focus and DU3 High Sensitivity Detector System. The CFI60 Apochromat Lambda S 40× water immersion objective lens, N.A. 1.25, W.D. 0.16–0.2 mm, F.O.V. 22 mm, DIC, correction collar 0.15–0.19 mm, spring-loaded was used for confocal image acquisition. The Kinetix CMOS camera and ×10 objective were used for bright field image acquisition. Nikon NIS-Element Package and ImageJ were used for image analysis.

### Cell quantification

Target cells in the dorsolateral, lateral, and dorsal regions of the subventricular zone (SVZ) were exhaustively quantified from five coronal sections per animal. Sections were spaced 300 μm apart, covering the entire SVZ. All cell counting was performed using ImageJ software.

### Fluorescent intensity and coverage measurement

Fluorescence intensity for MBP staining was quantified from three regions of interest (ROIs) in 4–5 coronal sections per animal. Area coverage was determined in ImageJ by calculating the proportion of signal above a predefined fluorescence threshold within the selected ROIs covering the corpus callosum.

### Statistical analysis for IHC

Sample sizes were chosen based on standard practices for reproducibility, recommendations from prior studies using the same knockout model, and the availability of KO animals; no formal statistical method was used to predetermine sample size. Genotypes were not blinded during experiments; however, microscopy images were blinded prior to cell counting and quantitative measurements. Student's $t$ test was used for comparisons between WT and KO groups at a single time point. For comparisons across multiple time points, a two-way ANOVA was performed. A $P$ value $< 0.05$ was considered statistically significant. When data did not meet normality assumptions, appropriate nonparametric tests were applied. All analyses were conducted in GraphPad Prism.

### SVZ tissue collection and single-cell dissociation

For single-cell experiments, P14 mice were anesthetized and hand perfused with ice-cold PBS. Brains were promptly harvested and placed in ice-cold HBSS without $Ca^{2+}$ and $Mg^{2+}$. The lateral wall of the subventricular zone (SVZ) was dissected following the protocol (Walker and Kempermann, 2014).

Tissue was dissociated using the Neural Tissue Dissociation Kit (P) (Miltenyi #130-092-628). Briefly, the dissected tissue was minced on ice using a #10 scalpel blade and transferred to enzyme mix 1. This mix consisted of buffer X, 70 μM beta-mercaptoethanol, 0.04% BSA, and enzyme P (added immediately before use). The tissue was incubated at 37 °C for 15 min with intermittent pipetting to aid in dissociation. Afterward, enzyme mix 2 (buffer Y and enzyme A) was added, and the tissue was incubated for an additional 10 min with further pipetting steps. Dissociation was stopped by adding HHB solution (HBSS containing calcium, magnesium, 10 mM HEPES, and 0.01% BSA), and the cell suspension was filtered through a 70-μm strainer.

The filtered cell suspension was centrifuged at $500 \times g$ for 10 min at 10 °C. Cells were resuspended in HHB and purified using a three-layer Percoll gradient (19%, 15%, and 11% Percoll in HBSS/HEPES). Following centrifugation at $430 \times g$ for 3 min at 4 °C, the top debris layer was discarded, and the remaining cells were pelleted, washed, and filtered through a 30-μm strainer. Cell counts were performed using trypan blue, and the concentration was adjusted to 5000 live cells/μL in cell suspension buffer provided by PIPseq (Fluent Biosciences) for subsequent cell lysis and library preparation as per the manufacturer's instructions.

### Cell preparation

Single-cell suspensions were prepared and diluted to 5000 cells/μL in Cell Suspension Buffer (Fluent Biosciences, FB0002440), with a total of 40,000 cells per reaction. Each suspension was supplemented with 40 U of RNase inhibitor (Millipore Sigma #3335399001).

### Capture and lysis

T20 PIPs (FB0003914) were thawed, briefly centrifuged, and placed on ice. Each T20 PIPs (FB003914) tube received 8 μL of the cell suspension, 40 U of RNase inhibitor, and 1 mL of Partitioning Reagent (FB0003123). After mixing and vortexing, the excess Partitioning Reagent was removed, leaving the PIPs+cells. Chemical Lysis Buffer 3 (CLB3, FB0003910) was added to the PIP tubes, and the samples were incubated in preheated PIPseq™ Dry bath (FBS-SCR-PDB, 25 °C for 15 min- 37 °C for 45 min–25 °C for 10 min), to complete lysis.

### mRNA isolation

Following lysis, PIPs were processed with Breaking Buffer (FB0003128) and De-Partitioning Reagent (FB0002516). PIPs were washed three times with chilled 1× Washing Buffer (FB0003139) using centrifugation at $750 \times g$ for 2 min. The final PIP volume was normalized to 250 μL based on weight to ensure consistency across samples.

### cDNA synthesis

Reverse transcription was performed with RT Enzyme Mix (FB0002206), RT Additive Mix (FB0002205), and Template Switching Oligo (TSO, FB0003140) in a total reaction volume of 250 μL. Dry bath lid was set to 105 °C and the following program

was used for cDNA synthesis: 25 °C for 30 min, 42 °C for 90 min, and 85 °C for 10 min.

### cDNA amplification
Whole transcriptome amplification (WTA) was conducted using a master mix comprising 4× PCR Master Mix (FB0004644) and WTA Primer (FB0002006). Amplification reactions were cycled according to protocol, yielding amplified cDNA for downstream library preparation.

### cDNA isolation and QC
Amplified cDNA was isolated from PIPs using magnetic bead cleanup. The purified cDNA was quantified using a Qubit 1× dsDNA High Sensitivity Assay Kit (ThermoFisher, Q33230), and fragment size distribution was assessed using a BioAnalyzer High Sensitivity DNA Kit (Agilent, 5067-4626).

### Library preparation
Purified cDNA was processed through fragmentation, end-repair, adapter ligation, and indexed PCR using reagents from the PIPseq T20 3' RNA Kit. Cleanup was performed after each step, with a double-sided size selection following indexing PCR. Libraries were assessed for concentration and quality using Qubit and fragment analysis prior to sequencing. This workflow provided sequencing-ready libraries within 2 days, suitable for downstream gene expression analysis and single-cell clustering.

### Sequencing
Premade libraries were sent to MedGenome for sequencing on the NovaSeq X+ platform. Libraries were sequenced using 10B lanes with a target of 400 million paired-end reads per library and a sequencing depth of 20,000 reads per input cell.

### Data preprocessing
Barcodes identifying individual PIPs/cells were assigned to sequenced cDNA constructs using PIPseeker™ v3.3. PIPseeker first identified barcode combinations and molecular identifiers (MI) for each read before mapping with STAR. After gathering molecule information, reads were assigned to genes by aligning coordinates against the STAR reference, with duplicate reads sharing the same MI collapsed into single counts.

The resulting counts were organized into a sparse matrix, where each entry corresponded to a cell barcode and gene. Barcodes were categorized as either cell-containing PIPs or background PIPs using a barcode rank plot ("knee plot"), which orders barcodes by transcript count. Following cell calling, directories were created for each sensitivity level, containing text files with selected cell barcodes, knee plot images, and filtered count matrices for downstream analysis.

### Initial QC and clustering
To maximize cell inclusion, we used the filtered gene expression matrix generated with the lowest sensitivity threshold. Additional cell filtering criteria were applied based on mitochondrial genome content (<10%) and the number of detected features (>500 and <6000) using Seurat (v1.4). Data normalization, scaling, and variable feature selection were conducted using SCTransform, which effectively mitigates technical noise and accounts for differences in sequencing depth. Principal Component Analysis (PCA) was then performed for dimensionality reduction.

To address batch effects, harmony was applied to the normalized data, ensuring that the batch-corrected data maintained biological variance. The corrected dataset was visualized using Uniform Manifold Approximation and Projection (UMAP) for dimensionality reduction and clustering.

Cell clustering was performed using the FindNeighbors and FindClusters functions, with a resolution parameter set to 0.55. Cluster-enriched genes were identified using the FindAllMarkers function with parameters min.pct = 0.25 and logFC.threshold = log(1.2). Cluster identities were assigned by cross-referencing known cell markers from the literature. Heatmaps displaying the top five enriched genes per cluster were generated using the DoHeatmap function. Additionally, violin plots for known cell markers identified (Zywitza et al, 2018) were generated for further validation of cluster identities.

### Subclustering and gene set scoring
To identify subclusters within specific cell populations, we isolated neural stem cells (NSCs), transit-amplifying progenitors (TAPs), and neuroblasts from our dataset. Subset data were analyzed using Seurat by rerunning the FindClusters function on principal components, with the resolution parameter set to 0.4. The results were visualized using t-distributed Stochastic Neighbor Embedding (t-SNE).

For cell type characterization and comparison with previously published data, we utilized gene sets from (Llorens-Bobadilla et al, 2015; Yang et al, 2023). Cells were scored using the AddModuleScore function for genes from the downloaded sets that were expressed in more than 50 UMIs across all cells of interest. The expression patterns of the gene set scores were visualized using violin plots.

### Differential gene expression analysis
To identify dysregulated genes in cells of interest derived from the Ndufs4 KO compared to the WT, we performed differential gene expression (DE) analysis for each cell cluster. The analysis was conducted using the FindAllMarkers function (Wilcoxon rank-sum test) in Seurat and edgeR (Robinson et al, 2010) for more robust statistical evaluation. Each cell was treated as an individual sample, and both replicates were analyzed independently (WT2 vs. NKO1 and WT1 vs. NKO2). Within each cluster, we compared gene expressions between conditions (KO vs. WT) to identify differentially expressed genes. We report genes that met the following criteria: (i) adjusted $P$ value < 0.05, (ii) log2 fold change ≥ +0.25 or ≤ −0.25, and (iii) consistent directionality of regulation across both replicates. All non-coding RNAs were removed from the analysis. The results were visualized using volcano plots, where the $x$ axis represents the log2 fold change, and the $y$ axis represents the −log10 (adjusted $P$ value.) Significant genes were highlighted to illustrate key findings in the data.

### Genetic ontology and gene network enrichment analysis
Functional enrichment analysis using DAVID was used to identify altered biological processes. Genes with an adjusted $P$ value < 0.05 and a log2 fold change ≥ +0.25 or ≤ −0.25 were included in the analysis. Additionally, DAVID was used to retrieve corresponding

EnsemblIDs for gene identifiers. For visualization, we utilized the GOplot R package to generate a circle plot, highlighting the top Gene Ontology (GO) terms. This visualization integrates fold-change data with GO enrichment results, providing insights into the direction of regulation (upregulated or downregulated) and the associated biological processes.

### Cluster proportion comparison analysis

The proportions of each cluster and subcluster were compared between groups using the scCODA v0.1.9 Python package (https://sccoda.readthedocs.io/en/latest/compositional_data.html). Ependymal cells were chosen as the reference group for cluster analysis, while for subcluster analysis, the reference cell type was automatically assigned by the software.

### Cell–cell communication analysis

CellChat was used to identify altered signaling pathways in each replicate individually by analyzing the information flow of each pathway (Jin et al, 2025). The information flow is calculated as the total communication probability between all pairs of cell groups within the inferred network (https://github.com/sqjin/CellChat).

### Proliferation scoring

Proliferation scoring was performed using the AddModuleScore function (Nestorowa et al, 2016), analogous to gene set scoring, but focused on genes associated with different phases of proliferation. For each cell cluster and condition, the cumulative fraction of cells was calculated against the proliferation score. To assess statistical significance, the Kolmogorov-Smirnov test was applied to compare the cumulative distributions between the two conditions for each cluster.

### Ingenuity pathway analysis (IPA)

DEGs from NDUFS4 KO NSCs, determined using the thresholds above, were submitted through Ingenuity Pathway Analysis software (Qiagen) for analyses of upstream regulators and identification of affected canonical pathways. The top ten inhibited and top ten activated upstream regulators were used to generate the associated bar plot. Neurodevelopmental disorders were selected from predicted diseases and functions output, and visually reconstructed in BioRender. From affected canonical pathways, oxidative phosphorylation was selected, and the expression level of respiratory chain complexes was presented using BioRender.

### Neurosphere isolation and assay

To grow neurospheres, dissociated cells from the lateral wall were harvested and cultured in DMEM/F12 medium (Gibco) supplemented with N2 supplement (Fisher Scientific #17502048), EGF (20 ng/μL, R&D Systems # 236-EG-200), and FGF (20 ng/μL, R&D Systems # 233-FB-025/CF) from P14 animals. Cells were plated in ultra-low adherent 24-well culture plates (Fisher Scientific #NC1882826), and half of the media was replaced every other day during a one-week incubation period. After 1 week, primary neurospheres were dissociated using Accutase (STEMCELL Technologies #07920), then replated in ultra-low adherent six-well plates for secondary neurosphere formation. After 3 days, neurospheres were transferred onto 2-chambered slides, and bright-field images were captured for analysis. The diameters of 50

**The paper explained**

**Problem**

Leigh syndrome (LS) is the most frequent pediatric mitochondrial disease, where neurodegenerative lesions in multiple brain regions result in high mortality for patients. Most phenotypes of the disease are easily explained by these lesions in areas like the brainstem and cerebellum, but often neurodevelopmental delays are also reported in these patients prior to degenerative phenotypes. The cellular mechanisms for these neurodevelopmental issues are not well studied or understood.

**Results**

In our study, we evaluated postnatal neurodevelopment of a well-established LS mouse model lacking a gene that is mutated in LS patients. We performed our evaluations of the brain at three time points before the age-of-onset when the mice also form these degenerative lesions. Neural stem progenitor cells (NSPCs) in the neurogenic niche called the subventricular zone (SVZ) were unable to proliferate, self-renew, or differentiate into other brain cell types such as neurons. The myelin-producing cells to facilitate neurotransmission were also disrupted in LS mice, who had thinner white matter tracts that connect the brain's hemispheres.

**Impact**

Magnetic resonance imaging for pediatric mitochondrial disease patients may miss subtle neurodevelopmental abnormalities. Considering that developmental delays are a feature of many other mitochondrial disorders, understanding the mechanisms behind these phenotypes is warranted.

neurospheres per replicate (three replicates per genotype) were measured using ImageJ. To compare the distribution of neurosphere diameters between WT and NDUFS4 KO, we performed an ordinal logistic regression analysis with a likelihood ratio test (LRT) in R.

### Graphics

Figures 5F, 6A, Appendix S2, and the synopsis were generated with Biorender.com. Figure EV1A was generated with Procreate.

## Data availability

Single-cell RNA-seq data have been deposited at GEO at GEO: GSE297078. All original code for downstream analysis has been deposited at Zenodo (https://doi.org/10.5281/zenodo.15424859). Full-resolution half-hemisphere microscopy images from P14 mouse brains corresponding to Fig. 1D have been deposited in the BioImage Archive (https://www.ebi.ac.uk/biostudies/bioimages/studies/S-BIAD2349).

The source data of this paper are collected in the following database record: biostudies:S-SCDT-10_1038-S44321-025-00367-4.

## Peer review information

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

## Acknowledgements

We thank the National Institutes of Health R01ES035013 (PDM) and R35GM142368 (AMP) for supporting this work.

## Author contributions

**Sahitya Ranjan Biswas**: Formal analysis; Investigation; Methodology; Writing—original draft; Writing—review and editing. **Porter L Tomsick**: Investigation; Visualization. **Colin Kelly**: Investigation; Visualization. **Brooke A Lester**: Investigation; Visualization. **Julia P Milner**: Investigation; Visualization. **Sara N Henry**: Investigation. **Yairis Soto**: Investigation. **Samantha Brindley**: Investigation. **Nicole DeFoor**: Investigation. **Paul D Morton**: Conceptualization; Formal analysis; Supervision; Funding acquisition; Visualization; Methodology; Writing—original draft; Project administration; Writing—review and editing. **Alicia M Pickrell**: Conceptualization; Formal analysis; Supervision; Funding acquisition; Investigation; Visualization; Writing—original draft; Project administration; Writing—review and editing.

Source data underlying figure panels in this paper may have individual authorship assigned. Where available, figure panel/source data authorship is listed in the following database record: biostudies:S-SCDT-10_1038-S44321-025-00367-4.

## Disclosure and competing interests statement

The authors declare no competing interests.

# Expanded View Figures

**Figure EV1.   Neural stem and progenitor cell spatial distribution between genotypes.**

(**A**) Graphical depiction of brain regions analyzed in this study generated with Procreate software. The region of interest is colored in blue. (**B–D**) Bregma distribution of SOX2[+] cells between WT and NDUFS4 KO mice in the SVZ at (**B**) P14, (**C**) P24, and (**D**) P30 ($n = 5$ mice/group). (**B**) $P = 0.2728$ (1.53), 0.9332 (1.23), 0.0018 (0.93), 0.0002 (0.63), <0.0001 (0.33). (**E, F**) Bregma distribution of SOX[+] DCX[+] cells between WT and NDUFS4 KO mice in the SVZ at (**E**) P24, (**F**) P30 ($n = 5$ mice/group). (**E**) $P = 0.1531$ (1.53), 0.0047 (1.23), 0.5558 (0.93), 0.7651 (0.63), 0.8901 (0.33). (**F**) $P = 0.9211$ (1.53), 0.0188 (1.23), 0.0519 (0.93), 0.0945 (0.63), 0.1319 (0.33). (**G–I**) Number of DCX[+] cells between WT and NDUFS4 KO mice in the SVZ at (**G**) P14, (**H**) P24, and (**I**) P30 ($n = 5$ mice/group). (**G**) $P = <0.0001$ (1.53), 0.0011 (1.23), 0.0497 (0.93), 0.4618 (0.63), 0.9992 (0.33). In (**B–I**), data represents mean and standard error of mean. Each dot represents one animal. *$P < 0.05$, **$P < 0.01$, ***$P < 0.001$, ****$P < 0.0001$, ns not significant (two-way ANOVA).

▶

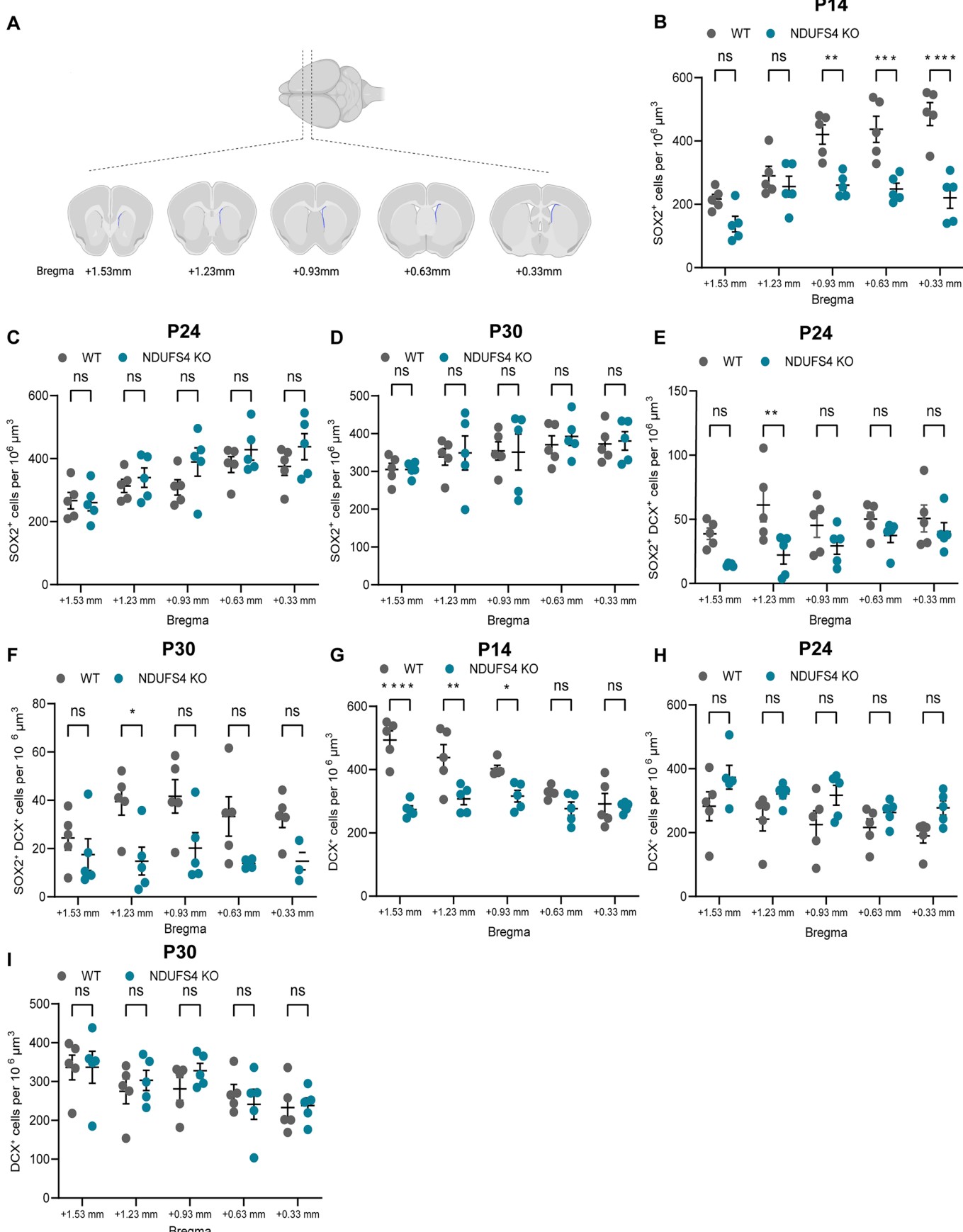

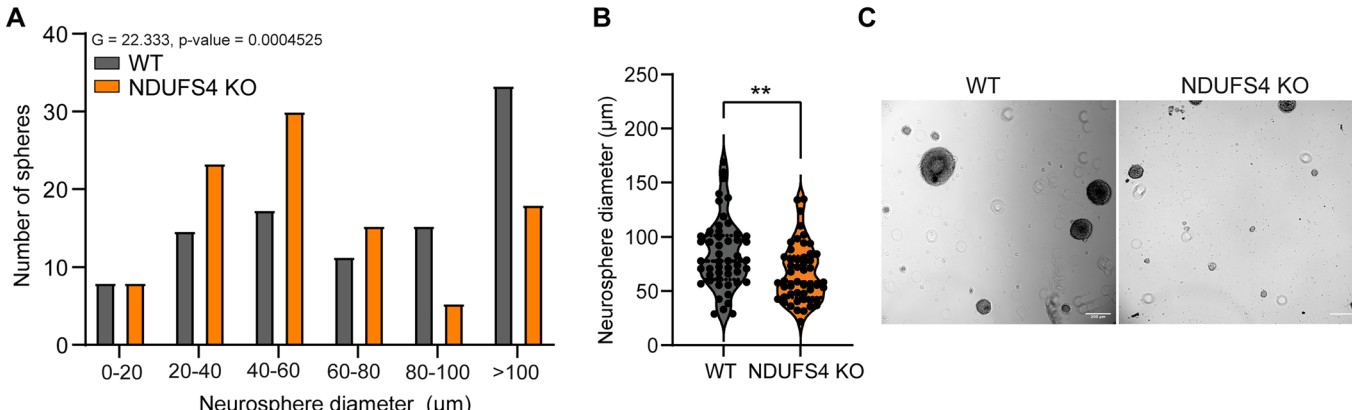

**Figure EV2. NDUFS4 KO neural stem cells display a reduced ability to proliferate in vitro.**

(A) Distribution of neurosphere diameter between WT and NDUFS4 KO animals. Neural stem cells were collected from 3 cultures derived from one animal/genotype. 50 neurospheres were measured per culture. Likelihood ratio test (LRT) was performed to compare the distribution. (B) Average neurosphere diameter between WT and NDUFS4 KO. $n = 150$ cells/group. $P = 0.0012$. Each dot represents one neurosphere. **$P < 0.01$ (Student's $t$ test). (C) Representative bright-field images for neurospheres. Scale bar $= 200 \, \mu m$.

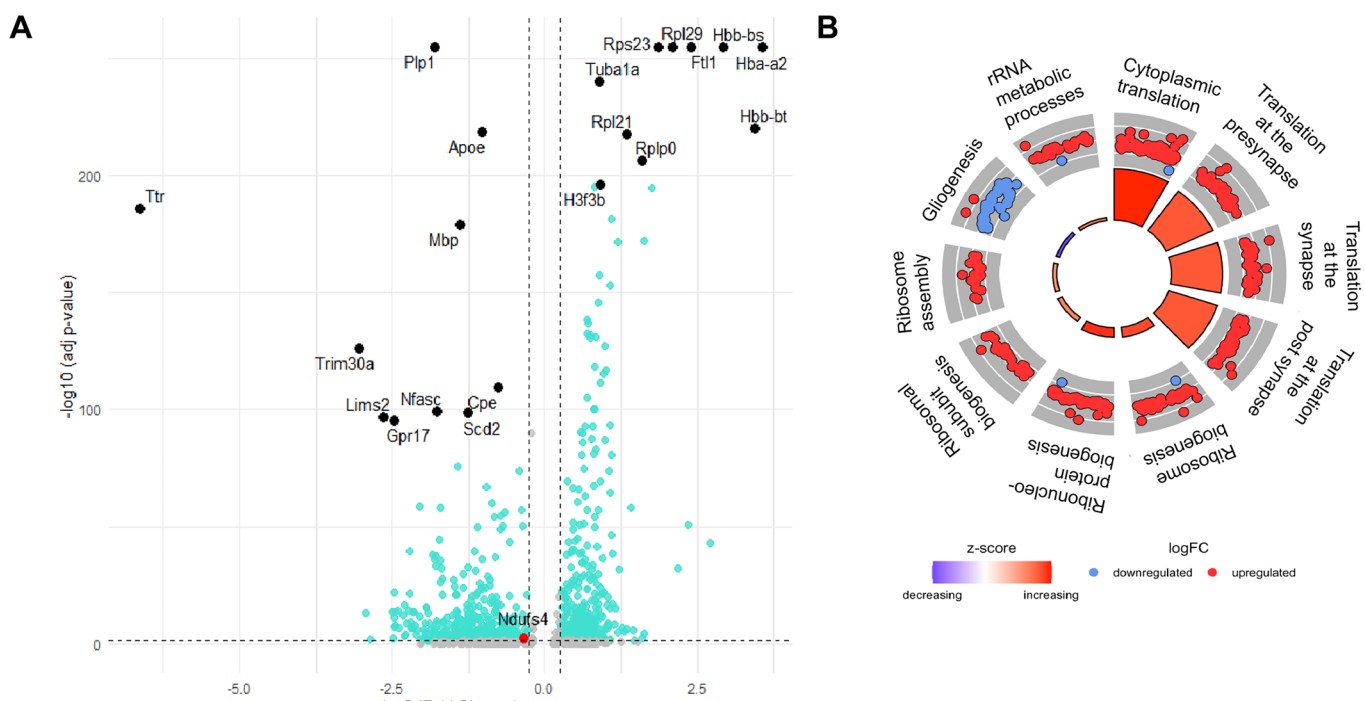

**Figure EV3.  Upregulated protein translation pathways in NDUFS4 KO neuroblasts.**

(**A**) Volcano plot showing fold change of differentially expressed genes in NDUFS4 KO neuroblasts. ($n = 1$ per group, then matched against the other replicate. Ones that are in the same direction were selected). Top 10 up and downregulated mRNAs are labeled (Black circles) and NDUFS4 is highlighted in the red circle. The threshold was set at log2(Fold Change) $\geq +0.25$ and $\leq -0.25$, and –log10 (adj $P$ value) $< 1.2991$. (**B**) GO circle plot showing top dysregulated biological processes and regulation of associated genes in NDUFS4 KO neuroblasts. In (**A**), Wilcoxon rank-sum test (nonparametric) was performed.

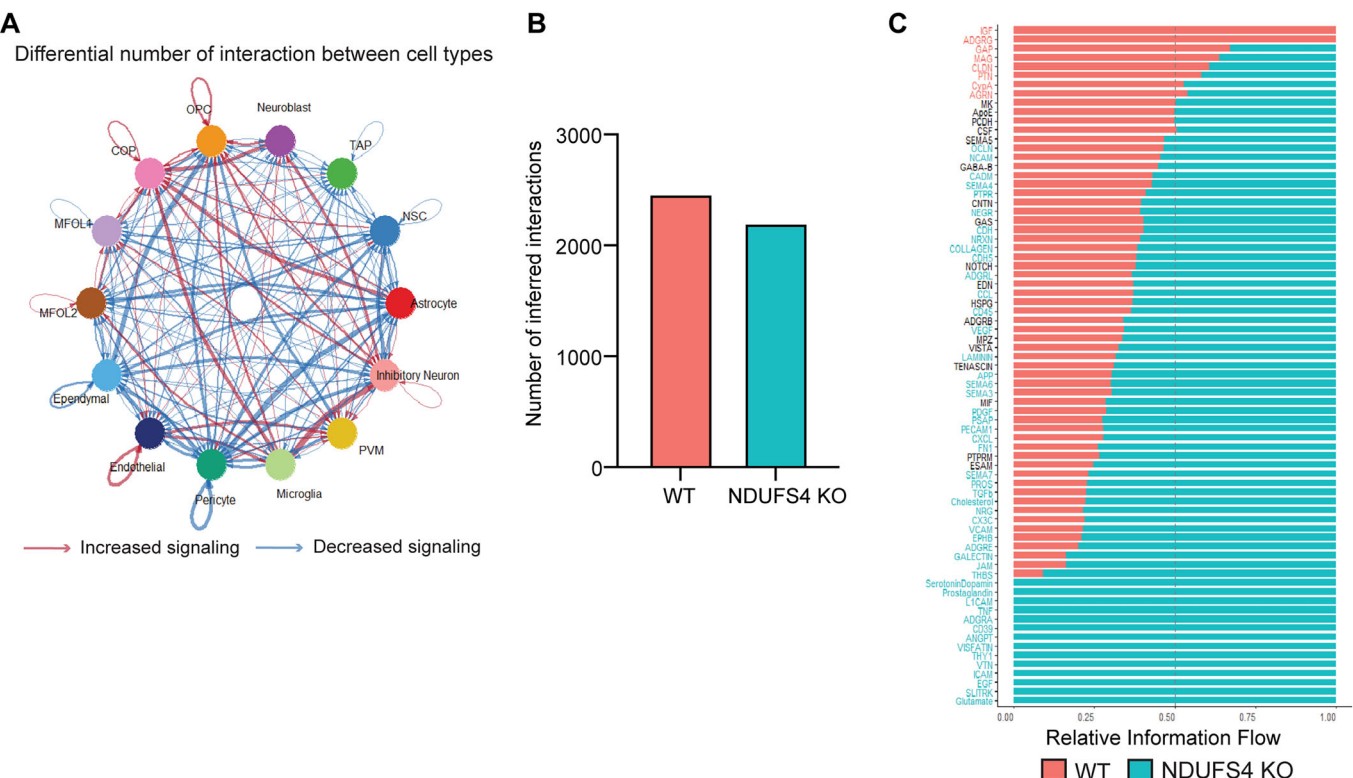

**Figure EV4. The number of signaling interactions is decreased in most cell types including neural progenitors in NDUFS4 KO as compared to WT.**

(A) Differential number of interactions between cell types from NDUFS4 KO. Blue and red lines indicate reduced and increased interactions between cells, respectively. (B) Total number of signaling interactions in all cells. (C) The significant signaling pathways were ranked based on their differences of overall information flow within the inferred networks between WT and NDUFS4 KO. The overall information flow of a signaling network is calculated by summarizing all the communication probabilities in that network. The top signaling pathways colored by red are more enriched in WT, and the bottom ones colored by cyan were more enriched in the NDUFS4 KO.

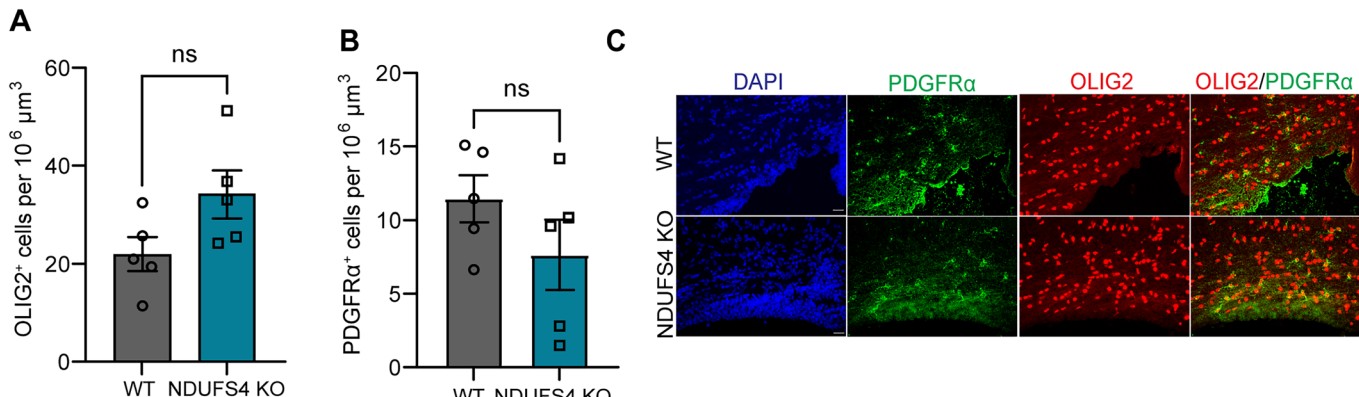

**Figure EV5. Oligodendrocytes are not changed in NDUFS4 KO SVZ at early postnatal day P14.**

(A, B) Density of (A) oligodendrocytes (OLIG2+) and (B) oligodendrocyte progenitors (PDGFRα+) at P14 ($n = 5$ mice/group). (C) Representative confocal images of immunohistochemical detection of PDGFRα (green), OLIG2 (red), and DAPI (blue). Scale bar = 20 μm. In (A, B), bar plot represents mean and standard error of mean. Each dot represents one animal. ns not significant (unpaired *t* test).

