## [Peer Review File · EMBO Molecular Medicine]

Neural/glial precursors and corpus callosum development postnatal defects in Leigh Syndrome mice

Sahitya Biswas, Porter Tomsick, Colin Kelly, Brooke Lester, Julia Milner, Sara Henry, Yairis Soto, Samantha Brindley, Nicole DeFoor, Paul Morton, and Alicia Pickrell

Corresponding authors: Alicia Pickrell (alicia.pickrell@vt.edu) , Paul Morton (pmorton@vt.edu)

Review Timeline:

Submission Date:	9th Jun 25
Editorial Decision:	21st Jul 25
Revision Received:	22nd Oct 25
Editorial Decision:	5th Dec 25
Revision Received:	8th Dec 25
Accepted:	10th Dec 25

Editor: Jingyi Hou

Transaction Report:

22nd Jul 2025

Dear Prof. Pickrell,

Thank you again for submitting your work to EMBO Molecular Medicine. We have now received the reports from the three reviewers and as you will see below, the reviewers think that the study is interesting and relevant. They raise however a series of concerns, which we would ask you to convincingly address in a revision.

The reviewers' recommendations are generally clear, so I won't reiterate them here. Regarding the brain morphological analysis before and after the first respiratory acts, as suggested by Referee #1, we will encourage you to pursue this, but it will not be a requirement for publication. All other issues raised need to be carefully addressed.

We would welcome the submission of a revised version within three months for further consideration. As you may already know, our editorial policy allows in principle a single round of major revision, and it is therefore essential to provide responses to the referees' comments that are as complete as possible.

Please also contact us as soon as possible if similar work is published elsewhere. If other work is published, we may not be able to extend the revision period beyond three months.

I look forward to receiving your revised manuscript.

Kind regards,
Jingyi

Jingyi Hou
Senior Editor
EMBO Molecular Medicine

We require:

2) Individual production quality figure files as .eps, .tif, .jpg (one file per figure). For guidance, download the 'Figure Guide PDF': (<https://www.embopress.org/page/journal/17574684/authorguide#figureformat>).

3) A .docx formatted letter INCLUDING the reviewers' reports and your detailed point-by-point responses to their comments. As part of the EMBO Press transparent editorial process, the point-by-point response is part of the Review Process File (RPF), which will be published alongside your paper.

4) A complete author checklist, which you can download from our author guidelines (<https://www.embopress.org/page/journal/17574684/authorguide#submissionofrevisions>). Please insert information in the checklist that is also reflected in the manuscript. The completed author checklist will also be part of the RPF.

6) It is mandatory to include a 'Data Availability' section after the Materials and Methods. Before submitting your revision, primary datasets produced in this study need to be deposited in an appropriate public database, and the accession numbers and database listed under 'Data Availability'. Please remember to provide a reviewer password if the datasets are not yet public (see <https://www.embopress.org/page/journal/17574684/authorguide#dataavailability>).

12) Author contributions: You will be asked to provide CRediT (Contributor Role Taxonomy) terms in the submission system. These replace a narrative author contribution section in the manuscript.

13) A Conflict of Interest statement should be provided in the main text.

14) Please provide a 'Synopsis' to further enhance discoverability. Synopses are displayed on the journal webpage and are

freely accessible to all readers. They include a short stand first (maximum of 300 characters, including space) as well as 2-5 one-sentences bullet points that summarizes the paper. Please write the bullet points to summarize the key NEW findings. They should be designed to be complementary to the abstract - i.e. not repeat the same text. We encourage inclusion of key acronyms and quantitative information (maximum of 30 words / bullet point). Please use the passive voice. Please attach these in a separate file or send them by email, we will incorporate them accordingly.

Please also suggest visual abstract to illustrate your article as a PNG file 550 px wide x 300-600 px high.

15) All Materials and Methods need to be described in the main text using our 'Structured Methods' format. According to this format, the Methods section includes a Reagents and Tools Table (listing key reagents, experimental models, software and relevant equipment and including their sources and relevant identifiers) followed by a Methods and Protocols section describing the methods, ideally using a step-by-step protocol format. The aim is to facilitate adoption of the methodologies across labs.

Please download and fill our Reagents and Tools Table template (.docx), which you can find in our author guidelines: <https://www.embopress.org/page/journal/17574684/authorguide#structuredmethods>

**** Reviewer's comments ****

Referee #1 (Comments on Novelty/Model System for Author):

The brain lesions of the NDUFS4 KO mouse model is not reproducing Leigh syndrome, which is a rather specific neuropathological entity. NDUFS4-KO brains more generically show diffuse spongiosis and neurodegeneration throughout the brain regions. The presence of presymptomatic lesions at birth and after a few hours of normal ventilation could clarify the role of high tension of oxygen in determining the onset of brain lesions that will eventually become associated with neurological symptoms in this interesting mouse model of mitochondrial driven early onset severe neurodegeneration.

Referee #1 (Remarks for Author):

In babies, Leigh syndrome is a neuropathological entity, first defined by Denis Leigh in 1951. It is characterized by necrotic lesions with neovascularization of the nuclei and surrounding white matter extending from the more rostral spinal metamers through the brainstem and dentate nuclei of cerebellum, up to the diencephalon (e.g. the Louis subthalamic nucleus) and basal ganglia (subacute encephalomyelopathy). In later stages cortical atrophy ensues with possibly some leukodystrophic lesions. Occasionally, a thin or posterior underdeveloped corpus callosum can be seen. A virtually invariant "disease free" window is observed at birth and can last from two to several months, before psychomotor stagnation and regression appear. The free window feature is a common characteristic, with few exceptions, of virtually all paediatric cases of mitochondrial disorders, suggesting that full introduction of air molecular oxygen by ventilation after birth can trigger the worsening of cerebral lesions until they become pathologically and clinically evident. The NDUFS4 KO mouse is not a faithful equivalent of human Leigh syndrome since the typical brainstem and basal ganglia symmetric necrotic lesions are missing, and the lesions are predominantly affecting the whole cerebral parenchyma with spongiosis, neuronal loss and cortical then global atrophy. The title should be changed accordingly (e.g. Early onset mitochondrial encephalopathy". Nevertheless, a disease free temporal window is also present in this model, which is therefore suitable to study the presymptomatic neuropathology in this model. The Authors did in fact found several presymptomatic lesions in the brain of this model, which show that progression of neuronal and glial failure precedes by some time the onset of the clinical manifestations. The description is crowded with acronyms that should be explained more explicitly, may be through a brief glossary. This is an interesting, albeit expected, findings, painstakingly described in the paper. However, the main question in this context is when these lesions begin, before or after birth? In the latter case it is likely that the trigger for the initiation of lesional alteration in the brain, may be the toxic effect of Oxygen-related ROS. Despite what has been suggested from data based in neuronal derivatives of NDUFS4 IPSC, Leigh disease in human and the NDUFS4 KO mouse do not display obvious malformative signs or evident arrest of neurodevelopment, with perhaps the exception of the aforementioned alterations in the corpus callosum. To establish whether the neurodegenerative process is triggered by the increase of O₂ tension due to normal pulmonary ventilation, I wonder whether it would be possible to extend the brain morphological analysis to brains of NDUFS4 and controls just before and just after the first respiratory acts, say in mice at the moment of expulsion from the parturition canal and after one or two hours (or one day) after birth. The verification of the presence or absence of significant lesions in this temporally well defined experimental procedure would add etio-pathological functional meaning to a paper that, in the present status, remains an excellent neuropathological observational but not functional investigation.

Referee #2 (Comments on Novelty/Model System for Author):

The data in this manuscript is high quality and well described, using a model organism that has been used extensively for modelling mitochondrial disease. It is notable that of the many publications that have used these mice, none have looked at phenotypes during postnatal development and therefore this manuscript is investigating a novel aspect of the disease. Further clarification in the text may help to better link their observations to the disease phenotypes.

Referee #2 (Remarks for Author):

This manuscript utilises a well-studied model of mitochondrial disease, the NDUFS4 knock-out mouse, to evaluate postnatal brain development and determine the contribution of developmental brain abnormalities to the disease phenotype. The authors identified hitherto undiscovered abnormalities during postnatal development in NDUFS4 knock-out mice brain morphology, as well as differences in neural stem cell populations and neuronal and oligodendrocyte lineage procession. These neurodevelopmental defects preceding the classic neurodegenerative phenotype observed in Leigh syndrome are interesting observations and uncovers contributions from neural progenitor cells to this process. The manuscript is clear and generally well written, provided interesting insight into the clinical features of Leigh Syndrome. It would be further strengthened by clarifying certain details and further exploring the differences in cell populations in the SVZ, as well as discussing the potential impacts in light of other conditional Ndufs4 knockouts that have been described.

1) Further clarity in the analysis of the neural stem cell progenitors and lineage progression in Fig. 2/Supp Fig.1, particularly across the regional assessments, would be helpful to strengthen the argument that the disruption to "lineage progression is evident at later ages" (p. 5 line 137). For instance, do the caudal changes observed in SOX2+ cells (Supp Fig 1B) at p14 normalise over time to account for the lack of overall insignificant differences at P24 and P30, or do some changes in the caudal regions persist? Likewise, the data in Supp Fig 1E at D30 shows pronounced differences of DCX+ cells in rostral regions, although Fig 2F only shows a significant difference at P14, and not P30? Are these regional differences in DCX+ cells more pronounced at P14?

As a general point, it would be helpful to indicate SVZ region in the schematic in Supp Fig 1A. As well, is the y axis label in Fig 3B correct? Should it be "SOX2+Ki67+/Ki67+" as the text appears to indicate (p. 6 lines 158-159)? Otherwise an explanation is required as to why cells undergoing transition (SOX2+DCX+) are being analysed here.

2) The difference in early versus late NBs (Fig 4G) is interesting, since it points to maturational biases in the neurogenic cells. It would also be of interest to look at global cell populations in the SVZ from sn-RNA seq data (Fig. 4A) to determine if there are other cell types that are altered and may contribute to the larger morphological changes? Global changes in cell populations here might support the altered gliogenesis pathways identified in the NBs and changes to oligodendrogenesis identified later. As a minor point, the abbreviations from Fig. 4A should be defined in the figure legend. These global changes might also help account for the size differences observed in the neurospheres (Supp Fig 2), although it would also be helpful to determine whether these size differences relate to total cell numbers, rather than cell type/size.

3) The decreased gliogenesis in the NSPCs is notable, and the authors have correlated this oligodendrocytic impairments through the decreased levels of MASH1+ progenitors and myelination in the corpus colosum. Considering that no evidence of demyelination was observed in a conditional knockout model lacking Ndufs4 in neurons and glia (Quintana et al. 2010, PMID: 20534480), and reduced myelin basic protein levels in the whole mouse Ndufs4 knockout only appearing to arise in older (p+30) mice (Johnson et al. 2021, PMID: 32331968), it would be of interest to determine if and how these changes persist over time? E.g. P14 versus P30

Likewise, how would the decreased gliogenesis in the NSPCs correlate with the gliosis observed in the Ndufs4 NesKO mice (Quintana et al. 2010, PMID: 20534480), when increased astrocytes would be expected?

Ultimately, the myelination defects and impacts on the corpus colosum are of interest to Leigh syndrome phenotypes, but it is still not totally clear how these correlate to the phenotypes of the conditional NDUFS4 knockouts, such as those described by Bolea et al (2019, PMID:31403401), and how much they contribute to the phenotype.

Minor issues to address:

Supp Fig 1 legend, line 53. Spelling mistake in "anova"

Fig 7C- is labelling on Y axis correct? I'm not sure why the graph is split?

P12 minor grammar mistakes in lines 345-346 "or other types of unphysiologically disruptions of mitochondrial function" and lines 350-351 "severely disrupts complex I cause embryonic lethality"

The sentences on P12 lines 350-354 (starting with "The loss of apoptosis-inducing factor.....") don't make sense with each other or in the context of mitochondrial function in stem cell biology.

The paragraph on p 12-13 lines 358-370 appears to have issues with referencing, and none of the references cited appear correct. For instance, the sentence on lines 362-365 "In another neurodevelopment disorder schizophrenia, the biggest genetic risk factor, 3q29Del, overwhelmingly points to alterations in mitochondrial DNA transcripts and nuclear genes that control OXPHOS in induced pluripotent stem cell (iPSCs) organoid and mouse models." cites an Alzheimer's paper rather than one on schizophrenia.

P 13 lines 367-368 "Altering the ability to generate new neurons development or even adulthood in the hippocampus and SVZ...." should be revised for clarity.

P17 line 512, there appears to be a mistake in the NDUFS4 mouse name (likely caused by conversion of the superscript letters).

Referee #3 (Comments on Novelty/Model System for Author):

The *Ndufs4* knockout mouse is the most robust LS model, showing early neurodevelopmental defects and progressive neurodegeneration, closely mimicking human disease. In contrast, *Surf1* knockout mice show mild or no spontaneous brain pathology. Conditional *Ndufs4* models allow cell-specific insights, while iPSC organoids offer human relevance but lack in vivo context. Overall, *Ndufs4* KO best balances translational relevance and mechanistic utility.

Referee #3 (Remarks for Author):

The manuscript "Impaired Complex I dysregulates neural/glia precursors and corpus callosum development, revealing postnatal defects in Leigh Syndrome mice" answers important questions about how mitochondrial dysfunction, especially Complex I impairment, affects neurodevelopment in Leigh Syndrome (LS), which has been considered a neurodegenerative disease. Using the NDUFS4 knockout (KO) mouse model, the authors aim to clarify whether defects in neural stem/progenitor cells (NSPCs) contribute to postnatal brain abnormalities. The study supports this central claim by integrating gross anatomical measurements, neurosphere-based proliferation assays, and single-cell transcriptomic profiling to demonstrate a marked decline in NSPC proliferation, neuroblast output, and oligodendroglial maturation in KO mice. A key claim—that Complex I dysfunction disrupts lineage progression—is substantiated by the identification of a maturational shift in neuroblast populations and downregulation of cell cycle genes in NSCs and transit-amplifying progenitors. Furthermore, region-specific analysis of the subventricular zone (SVZ) shows that progenitors derived from the medial and lateral ganglionic eminences are particularly vulnerable. Another major claim, that structural deficits such as reduced hemispheric width and corpus callosum abnormalities are linked to impaired neurogenesis and myelination, is well supported by decreased PLP expression and oligodendrocyte numbers. This study tackles an important and often overlooked angle of Leigh Syndrome by highlighting its neurodevelopmental aspects. The authors offer solid experimental evidence that early postnatal changes in neural and glial progenitor behavior could be driving some of the brain abnormalities seen in this mitochondrial disorder. However, the manuscript in its current form has several areas that need more clarity. Some key findings are underexplained, and the flow between sections could be improved. Most notably, the Discussion section needs thorough revision so it should better reflect and interpret the main results shown in the study. I've provided the following comments that I hope will help strengthen the manuscript and recommend a major revision.

Major:

1. The introduction section provides a fundamental overview of LS and its symptomatology; however, it should place greater attention on the NDUFS4 mutation spectrum and its effects on mitochondrial components and functionality. Additionally, incorporate mechanistic knowledge regarding how the mutation contributes to the neurodevelopmental phenotype.
2. There are two observations made on the neuroblast populations: (i) Lines 139-142 "Together, these results suggest a biphasic disruption in neurogenesis in complex I deficient mice. At P14, there were fewer neural stem/progenitor cells (Figure 2), leading to a reduced generation of neuroblasts, which may in part account for the maturational delays in brain growth exhibited (Figure 1). (ii) Lines 198-200: Further proportion analysis within the subclusters revealed a significant reduction in the percentage (33.88% vs. 1.75%) of cells identified as early NBs paired with an increase in late NBs (2.40% vs. 31.99%) in NDUFS4 KO compared with controls (Fig. 4E-G). The authors' claim in the initial statement regarding a maturation delay appears to contradict their subsequent observation, where the percentage of late-stage neuroblasts (NBs) is higher in NDUFS4 KO mice compared to wild-type controls. This finding suggests that early-stage NBs may be depleting more rapidly in the KO condition, potentially due to accelerated differentiation or exhaustion, rather than a delay in maturation. The authors should reconcile this discrepancy and clarify their interpretation of the data.
3. The differentially expressed genes (DEGs) identified in NSCs and TAPs, as shown in Figures A and B, are insufficiently interpreted and discussed in the Results section. For example, *CST3* is upregulated in NSCs, which is a gene with well-established roles in early brain patterning, homeostasis, and neuronal maturation (10.1073/pnas.0509789103). However, its relevance is not addressed in the manuscript. The authors should provide a more comprehensive interpretation of key DEGs, particularly those with known or plausible roles in neurodevelopment, to better support their proposed link between gene dysregulation and the observed neurodevelopmental deficits in the model.
4. Lines 243-245: "NDUFS4 KO TAPs and other cell types such as astrocytes had reduced interactions, resulting in changes in the overall information flow between cell types (Supplementary Fig. 6)." The results presented in Supplementary Figure 6a are addressed only superficially in the Results section. However, the figure suggests increased signaling interactions between neuroblasts and inhibitory neurons, as well as between COPs and neuroblasts, and microglia and inhibitory neurons. The authors should provide a more thorough interpretation of these observations and ensure that all interactions depicted in the figure are clearly described and discussed in the Results section.
5. In lines 245-247, the results section mentions only the role of NCAMs; however, Supplementary Figure 6C appears to show several other genes potentially involved in cell-cell interactions. The authors should consider expanding this section to acknowledge and interpret the broader range of genes presented, as this may strengthen the conclusions about neurodevelopmental signaling.
6. The discussion section, spanning pages 12 to 14 (lines 336-422), lacks a focused analysis of the study's actual findings. Upon

close reading, only lines 389-399 attempt to interpret the results, and even this portion is overly generalized and insufficient. The remainder of the discussion reads more like an extended introduction, with minimal engagement with the data. I recommend that the authors relocate most of the current content-excluding lines 389-399-to the introduction. The discussion should be comprehensively rewritten to critically examine and contextualize each of the results presented in the figures. It is particularly concerning that, while the results are well described and show clear interpretations, these are not adequately explored or integrated into the discussion.

Minor:

7. The manuscript inconsistently refers to the PLP1 gene as both PLP1 and PLP (e.g., Line 322). The nomenclature should be standardized throughout the text for clarity and consistency.

8. In several instances, entire figures are cited without specifying the relevant sub-panels-for example, Figure 1 (Line 142), Figure 2 (Lines 140 and 203), and Supplementary Figure 3 (Line 208). The authors are encouraged to reference specific sub-figures/panels where appropriate to precisely cite the particular result to readers.

On behalf of all the authors, we would like to thank the editors, editorial team, and the reviewers for spending their time and effort to read and prepare valuable comments for our manuscript.

Upon revision, we have prepared (i) 4 additional panels analyzing our dataset for Extended View Figure 1 and (ii) 10 new panels with 1 new experimental data set for Figure 7. We also rewrote portions of the introduction and the discussion of the manuscript as suggested by Reviewer 3. We have provided point-by-point responses for all comments and concerns raised by the three reviewers and have submitted all the source data and instructional checklists required by the editorial staff. We have added text in blue that addresses the reviewers' comments below and in the main manuscript. We greatly appreciate the opportunity to provide a revised, strengthened manuscript for review.

***** Reviewer's comments *****

Referee #1 (Comments on Novelty/Model System for Author):

The brain lesions of the NDUFS4 KO mouse model is not reproducing Leigh syndrome, which is a rather specific neuropathological entity. NDUFS4-KO brains more generically show diffuse spongiosis and neurodegeneration throughout the brain regions. The presence of presymptomatic lesions at birth and after a few hours of normal ventilation could clarify the role of high tension of oxygen in determining the onset of brain lesions that will eventually become associated with neurological symptoms in this interesting mouse model of mitochondrial driven early onset severe neurodegeneration.

Referee #1 (Remarks for Author):

In babies, Leigh syndrome is a neuropathological entity, first defined by Denis Leigh in 1951. It is characterized by necrotic lesions with neovascularization of the nuclei and surrounding white matter extending from the more rostral spinal metamers through the brainstem and dentate nuclei of cerebellum, up to the diencephalon (e.g. the Louis subthalamic nucleus) and basal ganglia (subacute encephalomyelopathy). In later stages cortical atrophy ensues with possibly some leukodystrophic lesions. Occasionally, a thin or posterior underdeveloped corpus callosum can be seen. A virtually invariant "disease free" window is observed at birth and can last from two to several months, before psychomotor stagnation and regression appear. The free window feature is a common characteristic, with few exceptions, of virtually all paediatric cases of mitochondrial disorders, suggesting that full introduction of air molecular oxygen by ventilation after birth can trigger the worsening of cerebral lesions until they become pathologically and clinically evident. The NDUFS4 KO mouse is not a faithful equivalent of human Leigh syndrome since the typical brainstem and basal ganglia symmetric necrotic lesions are missing, and the lesions are predominantly affecting the whole cerebral parenchyma with spongiosis, neuronal loss and cortical then global atrophy. The title should be changed accordingly (e.g. Early onset mitochondrial encephalopathy". Nevertheless, a disease free temporal window is also present in this model, which is therefore suitable to study the presymptomatic neuropathology in this model. The Authors did in fact find several presymptomatic lesions in the brain of this

model, which show that progression of neuronal and glial failure precedes by some time the onset of the clinical manifestations. The description is crowded with acronyms that should be explained more explicitly, may be through a brief glossary. This is an interesting, albeit expected, findings, painstakingly described in the paper. However, the main question in this context is when these lesions begin, before or after birth? In the latter case it is likely that the trigger for the initiation of lesional alteration in the brain, may be the toxic effect of Oxygen-related ROS. Despite what has been suggested from data based in neuronal derivatives of NDUFS4 IPSC, Leigh disease in human and the NDUFS4 KO mouse do not display obvious malformative signs or evident arrest of neurodevelopment, with perhaps the exception of the aforementioned alterations in the corpus callosum. To establish whether the neurodegenerative process is triggered by the increase of O₂ tension due to normal pulmonary ventilation, I wonder whether it would be possible to extend the brain morphological analysis to brains of NDUFS4 and controls just before and just after the first respiratory acts, say in mice at the moment of expulsion from the parturition canal and after one or two hours (or one day) after birth. The verification of the presence or absence of significant lesions in this temporally well defined experimental procedure would add etio-pathological functional meaning to a paper that, in the present status, remains an excellent neuropathological observational but not functional investigation.

We thank the Referee for their comprehensive review of our manuscript and for providing insightful interpretations to our findings. We appreciate the idea that initial ventilation following birth during a disease-free postnatal window may serve as a catalyst for disruption of neurodevelopmental processes whereby presymptomatic alterations may be detected in the brain. While the reviewer correctly defines which aspects of Leigh syndrome are faithfully recapitulated in NDUFS4-KO brains, studies indicate bilateral brainstem lesions visible by MRI in these mice during later stages (P50-60) of development during which hallmark neuropathological insults become evident (PMID: 28483998, PMID: 31402314). The reviewer raises a valid point regarding the potential inability to offload circulating oxygen bound to hemoglobin in the case of a Complex I deficiency resulting in parenchymal hyperoxia and ultimately cytotoxicity owing to ROS accumulation; while there is strong evidence for this phenomenon demonstrated by others in older >P30 NDUFS4 KO mice, this remains unexplored near birth when mammals transition from fetal hemoglobin to normal pulmonary ventilation following the first breath of *ex utero* life.

We agree with the reviewer that this increase in oxygen tension following natural birth may be the very onset of O₂ overaccumulation in the brain where mitochondria are incapable of proper O₂ consumption. The suggestion to assess significant lesions right before and after birth is well received and of high interest; however, there are several technical and logistical issues that pose major hindrances in exploring this line of inquiry without an additional study (bullet points below). In addition, we have no evidence of overt lesions at the ages assessed in our study and to capture diffuse injuries in such short peri-natal time windows when ROS levels may not have had adequate time to accumulate to toxic levels may be fruitless and would require many other experiments

to get to a functional level. We have substantially revised our Discussion section (lines 414-445) to include these important facets and the potential for changes in oxygen tension marked by birth as an underlying etiology of disease progression. In addition, we highlight planned experiments for future studies to provide more causal and functional insight into the understudied disease-free window in Leigh syndrome. Lastly, we discuss potential gas based and probe-based hypoxia therapies showing promise for disease outcomes in later stages of development that remain underexplored at very young ages in NDUFS4 KO mice.

Technical

- P0 pups are small and fragile. To generate enough tissue to test the outcome measures presented in our initial submission will require pooling together of several animal tissues; particularly harvesting enough cells for *in vitro* neurosphere assays and PIP-seq will merit an entire study of its own.
- To validate changes in brain, PO₂ would require administration of O₂ sensing probes, as P0 brains are likely too small for direct assessments with stereotaxic electrode implantations which require anesthesia (e.g. – new confound). In addition, we would need to validate region-specific changes throughout the brain as recent findings suggest that Complex I defects occur in a regional and temporal manner in these mice with CI activity declining with age (PMID: 26824698).
- In addition to validating hyperoxia in the brains of NDUFS4 KO newborns, we would need to assess O₂ in the blood along with hematocrit levels -which is technically challenging to collect- since we would be characterizing these metrics prior to the onset of respiratory impairments developed later in disease progression.

Logistical

- NDUFS4 KO mice are difficult to generate in high numbers with a probability of homozygosity of 1/16. In addition, we have found evidence of sexual dimorphisms in PLP as it is X-linked further complicating issues with adequate animal numbers (see below).

Confounding Interpretations

- OPCs are key regulators of angiogenesis in developing white matter tracts owing to hypoxia inducible factor (HIF1/2alpha) stabilization (PMID: 25018103). Recent genetic crossings with NDUFS4 KO mice to disrupt PHD oxygen sensing enzymes demonstrated that hypoxic breathing therapy to overcome hyperoxia in these mice is effective independent of HIF activation (PMID: 31402314). Therefore, in addition to use of O₂ sensing probes and regional assessments, we would need to evaluate OPCs apposed to, near, and far from blood vessels.

- Previous studies using hypoxia therapies in NDUFS4 KOs utilize mice at P30 to preclude any confounding factors related to hypoxia-induced vascular alterations documented in early development (PMID: 26917594, PMID: 28483998). A bonus at this postweaning age is lack of hypoxia-induced confounds from a foster mother rearing/feeding her pups in the chamber. Therefore, the key rescue experiment that would be required from such perinatal assessments would likely be limited to administrable small molecules to emulate hypoxia therapy to the pups which are relatively new, such as HypoxyStat (PMID: 39965572).

WM developmental timing issues

- One of our main findings was a sustained reduction in myelin proteins suggesting an inability for oligodendrocytes to myelinate axons. In humans, myelination begins in the 2nd-3rd trimester, whereas in mice myelination of the corpus callosum begins after the first postnatal week with the first myelin sheaths documented at P11 (PMID: 7453945). We see no loss in OPC numbers in the SVZ at P14 that would indicate cytotoxicity and determining the impact of first respiratory acts on this cell class may not be as developmentally aligned as the neural stem cell side of the study.

Referee #2 (Comments on Novelty/Model System for Author):

The data in this manuscript is high quality and well described, using a model organism that has been used extensively for modelling mitochondrial disease. It is notable that of the many publications that have used these mice, none have looked at phenotypes during postnatal development and therefore this manuscript is investigating a novel aspect of the disease. Further clarification in the text may help to better link their observations to the disease phenotypes.

Thank you for your positive comments, and we have reworked the text to better explain our observations in all aspects of the paper. We appreciate the time, effort, and thoughtful feedback provided by the reviewer. We have added new experiments and new data sets to address your questions.

Referee #2 (Remarks for Author):

This manuscript utilises a well-studied model of mitochondrial disease, the NDUFS4 knock-out mouse, to evaluate postnatal brain development and determine the contribution of developmental brain abnormalities to the disease phenotype. The authors identified hitherto undiscovered abnormalities during postnatal development in NDUFS4 knock-out mice brain morphology, as well as differences in neural stem cell populations and neuronal and oligodendrocyte lineage procession. These neurodevelopmental defects preceding the classic neurodegenerative phenotype observed in Leigh syndrome are interesting observations and uncovers contributions from neural progenitor cells to this process.

The manuscript is clear and generally well written, provided interesting insight into the clinical features of Leigh Syndrome. It would be further strengthened by clarifying certain details and further exploring the differences in cell populations in the SVZ, as well as discussing the potential impacts in light of other conditional *Ndufs4* knockouts that have been described.

1) Further clarity in the analysis of the neural stem cell progenitors and lineage progression in Fig. 2/Supp Fig.1, particularly across the regional assessments, would be helpful to strengthen the argument that the disruption to "lineage progression is evident at later ages" (p. 5 line 137). For instance, do the caudal changes observed in SOX2+ cells (Supp Fig 1B) at p14 normalise over time to account for the lack of overall insignificant differences at P24 and P30, or do some changes in the caudal regions persist?

Likewise, the data in Supp Fig 1E at D30 shows pronounced differences of DCX+ cells in rostral regions, although Fig 2F only shows a significant difference at P14, and not P30? Are these regional differences in DCX+ cells more pronounced at P14?

We agree with this line of questioning and added the bregma distribution for all the graphs that the reviewer is asking for in EV1 with 4 additional panels. In summary, SOX2⁺ cells display the defect in the caudal regions of the brain only at P14 (Figure EV1B-D) while the SOX2⁺DCX⁺ cells are only disrupted in the rostral area at the later time points (EV1E-F). DCX⁺ cells only display rostral defects at P14 and not later (EV1G-I). So, in short, no caudal defects occur past P14 from the markers that we observed.

As a general point, it would be helpful to indicate SVZ region in the schematic in Supp Fig 1A. As well, is the y axis label in Fig 3B correct? Should it be "SOX2+Ki67+/Ki67+" as the text appears to indicate (p. 6 lines 158-159)? Otherwise an explanation is required as to why cells undergoing transition (SOX2+DCX+) are being analysed here. We agree with the reviewer and have now colored/shaded the areas on the cartoon depictions of SVZ that we counted in blue shading in EV1.

Yes, we apologize. This is an error on our end have now fixed this y-axis on the figure.

2) The difference in early versus late NBs (Fig 4G) is interesting, since it points to maturational biases in the neurogenic cells. It would also be of interest to look at global cell populations in the SVZ from sn-RNA seq data (Fig. 4A) to determine if there are other cell types that are altered and may contribute to the larger morphological changes? Global changes in cell populations here might support the altered gliogenesis pathways identified in the NBs and changes to oligodendrogenesis identified later. As a minor point, the abbreviations from Fig. 4A should be defined in the figure legend.

We did cluster proportional analysis using scCODA for the main clusters in Figure 4A, but did not detect a significant difference. We have now written that no main cluster effect was found in the results on lines 197-199. We have also provided the abbreviations in the figure legend as suggested on lines 1065-1068.

These global changes might also help account for the size differences observed in the neurospheres (Supp Fig 2), although it would also be helpful to determine whether these size differences relate to total cell numbers, rather than cell type/size.

We attempted to compare total cell number, but postnatally NDUFS4 KO derived neurospheres were difficult to culture with an inability for neurospheres to continue past the first passage; hence, we analyzed the same number of WT and KO spheres' properties (lines 683-684). However, this section could be better described, so we have added lines 179-181 to the results to clarify.

3) The decreased gliogenesis in the NSPCs is notable, and the authors have correlated this oligodendrocytic impairments through the decreased levels of MASH1+ progenitors and myelination in the corpus collosum. Considering that no evidence of demyelination was observed in a conditional knockout model lacking *Ndufs4* in neurons and glia (Quintana et al. 2010, PMID: 20534480), and reduced myelin basic protein levels in the whole mouse *Ndus4* knockout only appearing to arise in older (p+30) mice (Johnson et al. 2021, PMID: 32331968), it would be of interest to determine if and how these changes persist over time? E.g. P14 versus P30

We agree with this reviewer and have performed the experiments to see if CC defects persist at the P30 time point. We found that both CC thickness and MBP staining were significantly decreased at P30 (Figure 7K-7Q). We have discussed this new finding in lines 349-351.

In reference to the papers cited by the reviewer, the discrepancy of the Nestin-cre conditional *Ndufs4* animals not displaying demyelination may be due to the evaluation of the cerebellum and not the corpus callosum in that manuscript, whereas Johnson et al. 2021 used western blotting for whole brain lysate at \geq P30 animals.

We removed PLP staining and the confocal imaging dataset from the first submission because we realized that as an X-linked gene we saw variation in the sexes for both WT and KO genotypes despite displaying significance in the first submission. All our datasets are mixed sex because the NDUFS4+/- X NDUFS4+/- matings required already has experimental animals generated at 1/16. We did not have the number of mice required to segregate sex for PLP staining at the later timepoint because we collected many of our experimental animals at P14.

Likewise, how would the decreased gliogenesis in the NSPCs correlate with the gliosis observed in the *Ndufs4* NesKO mice (Quintana et al. 2010, PMID: 20534480), when increased astrocytes would be expected?

The gliosis with microglia and astrocytes referred to in Quintana et al. 2010 was only found in the olfactory bulb and vestibular formation in early pre-symptomatic stages, only spreading to the rest of the brain in KO mice >40 days. Astrocytes can also induce proliferation themselves and reenter cell cycle without being derived from NSPCs upon injury (PMID: 18404517). We were interested in this experiment and was hoping to use our neurosphere cultures to examine differentiation potential, but technically this was

not feasible (see above). For us to properly examine this questions, we would need to pinpoint at what time point neuroinflammation is evident in the SVZ/CC region and examine astrocyte morphology and number before and after this time point, which would require the use of AAV viral vectors driven by astrocyte specific promoters for membrane tethered GFP or Tg animals crossed in NDUFS4 animals to accurately assess GFAP+ and GFAP- astrocytes (PMID: 30541903) alongside GFAP staining to exclude radial glia.

Ultimately, the myelination defects and impacts on the corpus collosum are of interest to Leigh syndrome phenotypes, but it is still not totally clear how these correlate to the phenotypes of the conditional NDUFS4 knockouts, such as those described by Bolea et al (2019, PMID:31403401), and how much they contribute to the phenotype.

The reference that the Reviewer 2 discusses is the generation of glutamatergic, GABAergic, and cholinergic specific NDUFS4 conditional KO animals. Considering that the CC is composed of axons derived from cortical neurons of Layer II/III and V which are mostly glutamatergic, the glutamatergic conditional KO mouse (which in that paper is most akin to the systemic KO animals) is most likely to display CC defects. We haven't ruled out whether what we are observing is CC dysmyelination or demyelination. This would require future studies where we examine even younger animals and perform electron microscopy to delineate these two possibilities. The glutamatergic cKO mice did display motor defects at the mid and late stages of life that were not reliant on problems with the neuromuscular junction, but whether these were related to CC demyelination or lesions in the cerebellum is not clear. To discuss this point, we have added lines 406-413 to the discussion.

Minor issues to address:

Supp Fig 1 legend, line 53. Spelling mistake in "anova"
This has now been corrected. That figure is now EV1.

Fig 7C- is labelling? I'm not sure why the graph is split?
We thank the reviewer for pointing this out. This was an error when generating the figure, and we have now unsplit the graph to remove the unnecessary axis break.

P12 minor grammar mistakes in lines 345-346 "or other types of unphysiologically disruptions of mitochondrial function" and lines 350-351 "severely disrupts complex I cause embryonic lethality"

We thank the reviewer for pointing out these grammatical and referencing errors in the discussion. We have rewritten the Discussion to this topic on lines 370-371.

The sentences on P12 lines 350-354 (starting with "The loss of apoptosis-inducing factor.....") don't make sense with each other or in the context of mitochondrial function in stem cell biology.

We agree and have omitted this line now.

The paragraph on p 12-13 lines 358-370 appears to have issues with referencing, and

none of the references cited appear correct. For instance, the sentence on lines 362-365 "In another neurodevelopment disorder schizophrenia, the biggest genetic risk factor, 3q29Del, overwhelmingly points to alterations in mitochondrial DNA transcripts and nuclear genes that control OXPHOS in induced pluripotent stem cell (iPSCs) organoid and mouse models." cites an Alzheimer's paper rather than one on schizophrenia.

We have removed this section of the Discussion as recommended by Reviewer 3.

P 13 lines 367-368 "Altering the ability to generate new neurons development or even adulthood in the hippocampus and SVZ...." should be revised for clarity.

See above.

P17 line 512, there appears to be a mistake in the NDUFS4 mouse name (likely caused by conversion of the superscript letters).

Thank you for pointing out this error. It has been corrected.

Referee #3 (Comments on Novelty/Model System for Author):

The Ndufs4 knockout mouse is the most robust LS model, showing early neurodevelopmental defects and progressive neurodegeneration, closely mimicking human disease. In contrast, Surf1 knockout mice show mild or no spontaneous brain pathology. Conditional Ndufs4 models allow cell-specific insights, while iPSC organoids offer human relevance but lack in vivo context. Overall, Ndufs4 KO best balances translational relevance and mechanistic utility.

We agree that it is a robust model of LS and our insights into postnatal development prior to symptomatic onset may shed light into neurodevelopmental defects described in these patients as well as provide additional insight into OXPHOS function in non-neuronal cell types in the brain that are often understudied.

Referee #3 (Remarks for Author):

The manuscript "Impaired Complex I dysregulates neural/glial precursors and corpus callosum development, revealing postnatal defects in Leigh Syndrome mice" answers important questions about how mitochondrial dysfunction, especially Complex I impairment, affects neurodevelopment in Leigh Syndrome (LS), which has been considered a neurodegenerative disease. Using the NDUFS4 knockout (KO) mouse model, the authors aim to clarify whether defects in neural stem/progenitor cells (NSPCs) contribute to postnatal brain abnormalities. The study supports this central claim by integrating gross anatomical measurements, neurosphere-based proliferation assays, and single-cell transcriptomic profiling to demonstrate a marked decline in NSPC proliferation, neuroblast output, and oligodendroglial maturation in KO mice. A key claim-that Complex I dysfunction disrupts lineage progression-is substantiated by the identification of a maturational shift in neuroblast populations and downregulation of cell cycle genes in NSCs and transit-amplifying progenitors. Furthermore, region-specific analysis of the subventricular zone (SVZ) shows that progenitors derived from

the medial and lateral ganglionic eminences are particularly vulnerable. Another major claim, that structural deficits such as reduced hemispheric width and corpus callosum abnormalities are linked to impaired neurogenesis and myelination, is well supported by decreased PLP expression and oligodendrocyte numbers.

This study tackles an important and often overlooked angle of Leigh Syndrome by highlighting its neurodevelopmental aspects. The authors offer solid experimental evidence that early postnatal changes in neural and glial progenitor behavior could be driving some of the brain abnormalities seen in this mitochondrial disorder. However, the manuscript in its current form has several areas that need more clarity. Some key findings are underexplained, and the flow between sections could be improved. Most notably, the Discussion section needs thorough revision so it should better reflect and interpret the main results shown in the study. I've provided the following comments that I hope will help strengthen the manuscript and recommend a major revision.

We thank this reviewer for their supportive comments on the findings and quality of the data. We find that the reviewer has made several strong points that we agree with where we could have better explained the results and in hindsight provided a better introduction and discussion. We have made changes to the writing of the manuscript as a whole and hope that the revisions have better highlighted the importance of this study overall.

Major:

1. The introduction section provides a fundamental overview of LS and its symptomatology; however, it should place greater attention on the NDUFS4 mutation spectrum and its effects on mitochondrial components and functionality. Additionally, incorporate mechanistic knowledge regarding how the mutation contributes to the neurodevelopmental phenotype.

We agree with the reviewer that there was a lack of information on the NDUFS4 mutation in the introduction and have added details to this section. Please see lines 85-93.

2. There are two observations made on the neuroblast populations: (i) Lines 139-142 "Together, these results suggest a biphasic disruption in neurogenesis in complex I deficient mice. At P14, there were fewer neural stem/progenitor cells (Figure 2), leading to a reduced generation of neuroblasts, which may in part account for the maturational delays in brain growth exhibited (Figure 1).

(ii) Lines 198-200: Further proportion analysis within the subclusters revealed a significant reduction in the percentage (33.88% vs. 1.75%) of cells identified as early NBs paired with an increase in late NBs (2.40% vs. 31.99%) in NDUFS4 KO compared with controls (Fig. 4E-G).

The authors' claim in the initial statement regarding a maturation delay appears to contradict their subsequent observation, where the percentage of late-stage neuroblasts (NBs) is higher in NDUFS4 KO mice compared to wild-type controls. This finding suggests that early-stage NBs may be depleting more rapidly in the KO condition, potentially due to accelerated differentiation or exhaustion, rather than a delay in maturation. The authors should reconcile this discrepancy and clarify their interpretation of the data.

We thank the reviewer for raising this point. In our scRNA-seq, we defined neuroblasts using a panel of genes as well as genes associated with lipid biosynthesis, glycolysis, cell cycle, ribosome, mitosis, and neuronal differentiation. Doublecortin (DCX) is known to decline as neuroblasts progress to late stages and upregulate neuronal genes (PMID: 14574675). Accordingly, the IHC in Fig. 2F primarily captures early-stage DCX⁺ neuroblasts, and the observed reduction reflects a loss of this population. We have clarified this in the results. Please see lines 208-209.

3. The differentially expressed genes (DEGs) identified in NSCs and TAPs, as shown in Figures A and B, are insufficiently interpreted and discussed in the Results section. For example, CST3 is upregulated in NSCs, which is a gene with well-established roles in early brain patterning, homeostasis, and neuronal maturation (10.1073/pnas.0509789103). However, its relevance is not addressed in the manuscript. The authors should provide a more comprehensive interpretation of key DEGs, particularly those with known or plausible roles in neurodevelopment, to better support their proposed link between gene dysregulation and the observed neurodevelopmental deficits in the model.

We agree with the Reviewer that we have not adequately described DEG changes in TAPs despite it being a main figure. We have now added additional information on lines 246-253 to point out specific DEG changes in TAPs, and have included CST3 as suggested. We overlooked CST3 initially because although it was significant, it was not a top dysregulated DEG, albeit not less important.

4. Lines 243-245: "NDUFS4 KO TAPs and other cell types such as astrocytes had reduced interactions, resulting in changes in the overall information flow between cell types (Supplementary Fig. 6)." The results presented in Supplementary Figure 6a are addressed only superficially in the Results section. However, the figure suggests increased signaling interactions between neuroblasts and inhibitory neurons, as well as between COPs and neuroblasts, and microglia and inhibitory neurons. The authors should provide a more thorough interpretation of these observations and ensure that all interactions depicted in the figure are clearly described and discussed in the Results section.

We agree with the reviewer that our description of cell-cell interaction data was limited. We have now addressed the interactions involving cells of both neurogenic and non-neurogenic lineages and broaden the interpretation beyond NCAMs to include KO-enriched signaling pathways which may have implications in neurogenesis. Please refer to lines 258-273.

5. In lines 245-247, the results section mentions only the role of NCAMs; however, Supplementary Figure 6C appears to show several other genes potentially involved in cell-cell interactions. The authors should consider expanding this section to acknowledge and interpret the broader range of genes presented, as this may strengthen the conclusions about neurodevelopmental signaling.

See above

6. The discussion section, spanning pages 12 to 14 (lines 336-422), lacks a focused

analysis of the study's actual findings. Upon close reading, only lines 389-399 attempt to interpret the results, and even this portion is overly generalized and insufficient. The remainder of the discussion reads more like an extended introduction, with minimal engagement with the data. I recommend that the authors relocate most of the current content-excluding lines 389-399-to the introduction. The discussion should be comprehensively rewritten to critically examine and contextualize each of the results presented in the figures. It is particularly concerning that, while the results are well described and show clear interpretations, these are not adequately explored or integrated into the discussion.

We have completely rewritten large portions of the Discussion as suggested to incorporate comments from Reviewer 1 and Reviewer 2. We reintroduced our discussion on what is known about mitochondrial biology and neural stem cells/stem cells and omitted the discussion related to neurodegenerative diseases that are inherited that may have a neurodevelopmental origin. We kept the section that Reviewer 3 did find acceptable and added to those points to address Reviewer 2.

Minor:

7. The manuscript inconsistently refers to the PLP1 gene as both PLP1 and PLP (e.g., Line 322). The nomenclature should be standardized throughout the text for clarity and consistency.

Please see Reviewer 2 concern 3. We have now replaced that dataset with a more thorough evaluation of MBP at two different ages in Figure 7.

8. In several instances, entire figures are cited without specifying the relevant sub-panels-for example, Figure 1 (Line 142), Figure 2 (Lines 140 and 203), and Supplementary Figure 3 (Line 208). The authors are encouraged to reference specific sub-figures/panels where appropriate to precisely cite the particular result to readers. We apologize for this oversight and have now reviewed the manuscript to ensure that specific subpanels are referred to throughout the manuscript.

5th Dec 2025

Dear Prof. Pickrell,

Thank you for sending us your revised manuscript. We have now heard back from the two reviewers who were asked to re-evaluate your study. As you will see, the reviewers are both satisfied with the modifications made. Before we can formally accept your manuscript, we would ask you to address the following editorial-level issues:

1. Please reduce the keyword number to five.
2. Please remove the "Authors' contribution" section.
3. The references need to be formatted according to the EMBO Molecular Medicine reference style. Please list up to 10 co-authors of a paper before adding et al. in the reference list. Citations should be listed in alphabetical order. Remove DOIs for published papers.
4. "Declaration of Interests" should be renamed to "Disclosure and competing interests statement".
5. Remove the list of abbreviations from the manuscript text and incorporate the abbreviations into the relevant sections of manuscript text.
6. BIORENDER: Please remove from figure legends and add a dedicated "graphics" section to the Methods, following this format:
Graphics:
(some of the... OR Figure #... OR synopsis) Graphics were created with BioRender.com.
7. The synopsis image should be submitted in PNG format rather than PDF, with the required dimensions of 550 px in width and 300-600 px in height.
8. Data availability
 - Please remove the reviewer access code for GSE297078 and make sure the dataset will be made publicly available upon the acceptance of the manuscript.
 - Please provide specific URL for GSE297078 in this section.
9. Rename Tables EV1 - EV3 to "Dataset EV1 - EV3". Remove the legends from the manuscript text.
10. Appendix: Please correct the nomenclature to "Appendix Figure Sx" in the legends and remove the blue font.
11. In the Author Checklist, under the Ethics section, please provide the missing information regarding animal experiments.
12. In the figure legends, correct the nomenclature of expanded view figures to "Figure EV X".
13. Please address the following issues in the figure legends:
 - Please note that the legend for figure 5 is not provided in a sequential manner. This needs to be rectified.
 - Please note that the exact p values are not provided in the legends of figures 1B, C; 2A, C, D, F; 3A, C; 6C-E; 7D, G, I, J, K, N, P, Q. S; EV1 B, E, F, G; 2B
 - Please indicate the statistical test used for data analysis in the legends of figures 5A, C; EV3 A
 - Please note that information related to n is missing in the legends of figures 5A, C; EV3 A

I look forward to reading a new revised version of your manuscript as soon as possible.

Kind regards,
Jingyi

Jingyi Hou
Senior Editor
EMBO Molecular Medicine

*** Instructions to submit your revised manuscript ***

When preparing your revised manuscript, please refer to our guidelines: <https://link.springer.com/journal/44321/submission-guidelines#cms-Revised-submissions>. We perform an initial quality control of all revised manuscripts before re-review; failure to include requested items will delay the evaluation of your revision.

***** Reviewer's comments *****

Referee #2 (Comments on Novelty/Model System for Author):

As with the initial submission, the data in this revised manuscript is high quality and well described. The model organism is well established and has been used extensively for modelling mitochondrial disease. This manuscript focuses on phenotypes during postnatal development in these mice, an area that has not been previously explored in this model. The revised manuscript provides important insight into disease progression in this mouse model, which is nicely linked to symptoms in Leigh Syndrome in humans.

Referee #2 (Remarks for Author):

The revised manuscript contains significant new data that has addressed all my concerns, and provides additional insight into neural progenitor cells and myelination in this model across development. As well, the revised discussion more clearly links the authors observations to disease progression and symptoms, providing better clarity around the impact of their findings. The insights into the clinical presentation of Leigh Syndrome are novel, of high interest, and identify cellular mechanisms that may explain some of the neurodevelopmental features of the disease. These observations may be relevant across other mitochondrial diseases, and in my opinion this revised manuscript now warrants publication. I do note however that there appeared to be some technical issues with Tables EV2 and EV3 in this version of the manuscript, which were missing in the version I downloaded. I do not believe that these tables have changed from the original submitted manuscript, so this did not impact my interpretation of the data. However, this will need to be rectified before publication.

Referee #3 (Comments on Novelty/Model System for Author):

The Ndufs4 knockout mouse is the most robust LS model, showing early neurodevelopmental defects and progressive neurodegeneration, closely mimicking human disease. In contrast, Surf1 knockout mice show mild or no spontaneous brain pathology. Conditional Ndufs4 models allow cell-specific insights, while iPSC organoids offer human relevance but lack in vivo context. Overall, Ndufs4 KO best balances translational relevance and mechanistic utility.

Referee #3 (Remarks for Author):

The authors have satisfactorily addressed all of my major and minor comments. The revised manuscript now provides clearer mechanistic context for NDUFS4 dysfunction, reconciles the neuroblast maturation interpretation, expands the analysis of key DEGs, and substantially improves the treatment of cell-cell interaction data. The Discussion section has been thoroughly rewritten and now appropriately contextualizes the findings within the study's core results. I recommend acceptance of this manuscript for publication.

The authors addressed the remaining editorial issues.

10th Dec 2025

Dear Prof. Pickrell,

We are pleased to inform you that your manuscript is accepted for publication and is now being sent to our publisher to be included in the next available issue of EMBO Molecular Medicine.

You may qualify for financial assistance for your publication charges - either via a Springer Nature fully open access agreement or an EMBO initiative. Check your eligibility: <https://link.springer.com/journal/44321/how-to-publish-with-us>

Sincerely,
Jingyi

Jingyi Hou
Senior Editor
EMBO Molecular Medicine

>>> Please note that it is EMBO Molecular Medicine policy for the transcript of the editorial process (containing referee reports and your response letter) to be published as an online supplement to each paper. If you do NOT want this, you will need to inform the Editorial Office via email immediately. More information is available here: <https://link.springer.com/partners/embo-press/editorial-policies#Peer%20review>